# Unsupervised State Representation Learning in Atari

**Ankesh Anand**[*]
Mila, Université de Montréal
Microsoft Research

**Evan Racah**[*]
Mila, Université de Montréal

**Sherjil Ozair**[*]
Mila, Université de Montréal

**Yoshua Bengio**
Mila, Université de Montréal

**Marc-Alexandre Côté**
Microsoft Research

**R Devon Hjelm**
Microsoft Research
Mila, Université de Montréal

## Abstract

State representation learning, or the ability to capture latent generative factors of an environment, is crucial for building intelligent agents that can perform a wide variety of tasks. Learning such representations without supervision from rewards is a challenging open problem. We introduce a method that learns state representations by maximizing mutual information across spatially and temporally distinct features of a neural encoder of the observations. We also introduce a new benchmark based on Atari 2600 games where we evaluate representations based on how well they capture the ground truth state variables. We believe this new framework for evaluating representation learning models will be crucial for future representation learning research. Finally, we compare our technique with other state-of-the-art generative and contrastive representation learning methods. The code associated with this work is available at
https://github.com/mila-iqia/atari-representation-learning

## 1 Introduction

The ability to perceive and represent visual sensory data into useful and concise descriptions is considered a fundamental cognitive capability in humans [1, 2], and thus crucial for building intelligent agents [3]. Representations that concisely capture the true state of the environment should empower agents to effectively transfer knowledge across different tasks in the environment, and enable learning with fewer interactions [4].

Recently, deep representation learning has led to tremendous progress in a variety of machine learning problems across numerous domains [5, 6, 7, 8, 9]. Typically, such representations are often learned via end-to-end learning using the signal from labels or rewards, which makes such techniques often very sample-inefficient. Human perception in the natural world, however, appears to require almost no explicit supervision [10].

Unsupervised [11, 12, 13] and self-supervised representation learning [14, 15, 16] have emerged as an alternative to supervised versions which can yield useful representations with reduced sample complexity. In the context of learning state representations [17], current unsupervised methods rely on generative decoding of the data using either VAEs [18, 19, 20, 21] or prediction in pixel-space [22, 23]. Since these objectives are based on reconstruction error in the pixel space, they are not incentivized to capture abstract latent factors and often default to capturing pixel level details.

In this work, we leverage recent advances in self-supervision that rely on scalable estimation of mutual information [24, 25, 26, 27], and propose a new contrastive state representation learning

---

[*]Equal contribution. {anandank, racaheva, ozairs}@mila.quebec

method named Spatiotemporal DeepInfomax (ST-DIM), which maximizes the mutual information across both the spatial and temporal axes.

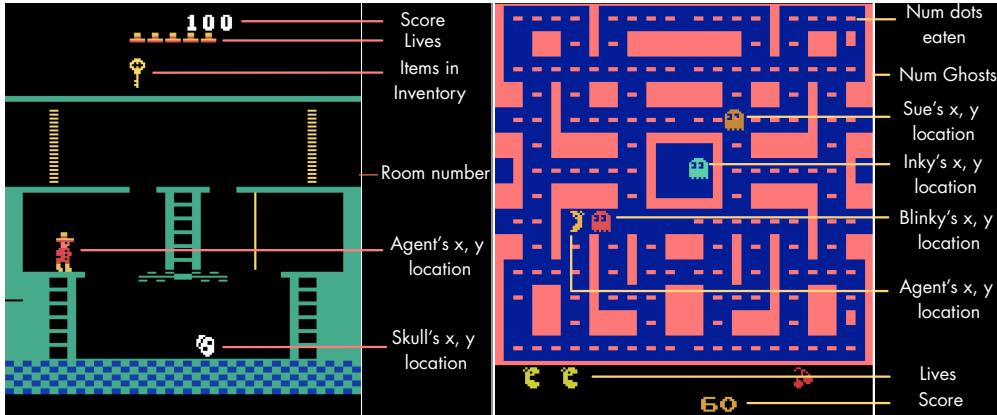

Figure 1: We use a collection of 22 Atari 2600 games to evaluate state representations. We leveraged the source code of the games to annotate the RAM states with important state variables such as the location of various objects in the game. We compare various unsupervised representation learning techniques based on how well the representations linearly-separate the state variables. Shown above are examples of state variables annotated for Montezuma's Revenge and MsPacman.

To systematically evaluate the ability of different representation learning methods at capturing the true underlying factors of variation, we propose a benchmark based on Atari 2600 games using the Arcade Learning Environment [ALE, 28]. A simulated environment provides access to the underlying generative factors of the data, which we extract using the source code of the games. These factors include variables such as the location of the player character, location of various items of interest (keys, doors, etc.), and various non-player characters, such as enemies (see figure 2). Performance of a representation learning technique in the Atari representation learning benchmark is then evaluated using *linear probing* [29], i.e. the accuracy of linear classifiers trained to predict the latent generative factors from the learned representations.

Our contributions are the following

1. We propose a new self-supervised state representation learning technique which exploits the spatial-temporal nature of visual observations in a reinforcement learning setting.
2. We propose a new state representation learning benchmark using 22 Atari 2600 games based on the Arcade Learning Environment (ALE).
3. We conduct extensive evaluations of existing representation learning techniques on the proposed benchmark and compare with our proposed method.

## 2 Spatiotemporal Deep Infomax

We assume a setting where an agent interacts with an environment and observes a set of high-dimensional observations $\mathcal{X} = \{x_1, x_2, \ldots, x_N\}$ across several episodes. Our goal is to learn an abstract representation of the observation that captures the underlying latent generative factors of the environment.

This representations should focus on high-level semantics (e.g., the concept of agents, enemies, objects, score, etc.) and ignore the low-level details such as the precise texture of the background, which warrants a departure from the class of methods that rely on a generative decoding of the full observation. Prior work in neuroscience [30, 31] has suggested that the brain maximizes *predictive information* [32] at an abstract level to avoid sensory overload. Predictive information, or the mutual information between consecutive states, has also been shown to be the organizing principle of retinal ganglion cells in salamander brains [33]. Thus our representation learning approach relies on maximizing an estimate based on a lower bound on the mutual information over consecutive observations $x_t$ and $x_{t+1}$.

## 2.1 Maximizing mutual information across space and time

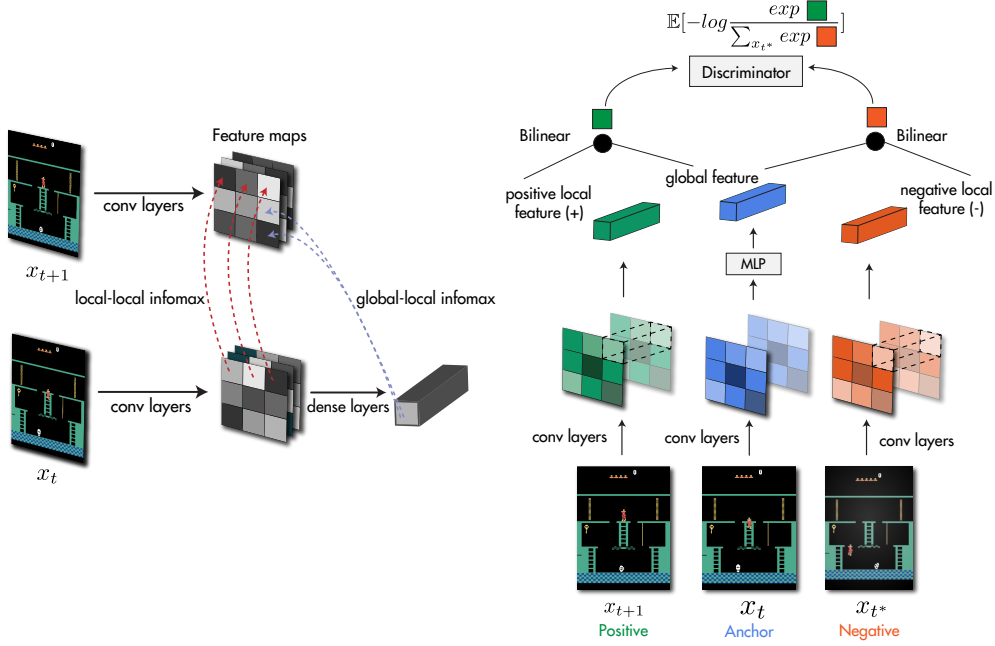

Figure 2: A schematic overview of SpatioTemporal DeepInfoMax (ST-DIM). Left: The two different mutual information objectives: local-local infomax and global-local infomax. Right: A simplified version of the global-local contrastive task. In practice, we use multiple negative samples.

Given a mutual information estimator, we follow DIM [26] and maximize a sum of patch-level mutual information objectives. The global-local objective in equation 2 maximize the mutual information between the full observation at time $t$ with small patches of the observation at time $t + 1$. The representations of the small image patches are taken to be the hidden activations of the convolutional encoder applied to the full observation. The layer is picked appropriately to ensure that the hidden activations only have a limited receptive field corresponding to $1/16^{th}$ the size of the full observations. The local-local objective in equation 3 maximizes the mutual information between the local feature at time $t$ with the corresponding local feature at time $t + 1$. Figure **??** is a visual depiction of our model which we call Spatiotemporal Deep Infomax (ST-DIM).

It has been shown that mutual information bounds can be loose for large values of the mutual information [34] and in practice fail to capture all the relevant features in the data [35] when used to learn representations. To alleviate this issue, our approach constructs multiple small mutual information objectives (rather than a single large one) which are easier to estimate via lower bounds, which has been concurrently found to work well in the context of semi-supervised learning [36].

For the mutual information estimator, we use infoNCE [25], a multi-sample variant of noise-contrastive estimation [37] that was also shown to work well with DIM. Let $\{(x_i, y_i)\}_{i=1}^N$ be a paired dataset of $N$ samples from some joint distribution $p(x, y)$. For any index $i$, $(x_i, y_i)$ is a sample from the joint $p(x, y)$ which we refer to as *positive examples*, and for any $i \neq j$, $(x_i, y_j)$ is a sample from the product of marginals $p(x)p(y)$, which we refer to as *negative examples*. The InfoNCE objective learns a score function $f(x, y)$ which assigns large values to positive examples and small values to negative examples by maximizing the following bound [see 25, 38, for more details on this bound],

$$\mathcal{I}_{NCE}(\{(x_i, y_i)\}_{i=1}^N) = \sum_{i=1}^{N} \log \frac{\exp f(x_i, y_i)}{\sum_{j=1}^{N} \exp f(x_i, y_j)} \qquad (1)$$

The above objective has also been referred to as *multi-class n-pair loss* [39, 40] and *ranking-based NCE* [41], and is similar to MINE [24] and the JSD-variant of DIM [26].

Following van den Oord et al. [25] we use a bilinear model for the score function $f(x,y) = \phi(x)^T W \phi(y)$, where $\phi$ is the representation encoder. The bilinear model combined with the InfoNCE objective forces the encoder to learn *linearly predictable* representations, which we believe helps in learning representations at the semantic level.

Let $X = \{(x_t, x_{t+1})_i\}_{i=1}^{B}$ be a minibatch of consecutive observations that are randomly sampled from several collected episodes. Let $X_{next} = X[:, 1]$ correspond to the set of next observations. In our context, the positive samples correspond to pairs of consecutive observations $(x_t, x_{t+1})$ and negative samples correspond to pairs of non-consecutive observations $(x_t, x_{t^*})$, where $x_{t^*}$ is a randomly sampled observation from the same minibatch.

As mentioned above, in ST-DIM, we construct two losses: the global-local objective (GL) and the local-local objective (LL). The global-local objective is as follows:

$$\mathcal{L}_{GL} = \sum_{m=1}^{M} \sum_{n=1}^{N} - \log \frac{\exp(g_{m,n}(x_t, x_{t+1}))}{\sum_{x_t^* \in X_{next}} \exp(g_{m,n}(x_t, x_{t^*}))} \tag{2}$$

where the score function for the global-local objective, $g_{m,n}(x_t, x_{t+1}) = \phi(x_t)^T W_g \phi_{m,n}(x_{t+1})$ and $\phi_{m,n}$ is the local feature vector produced by an intermediate layer in $\phi$ at the $(m, n)$ spatial location.

The local-local objective is as follows:

$$\mathcal{L}_{LL} = \sum_{m=1}^{M} \sum_{n=1}^{N} - \log \frac{\exp(f_{m,n}(x_t, x_{t+1}))}{\sum_{x_t^* \in X_{next}} \exp(f_{m,n}(x_t, x_{t^*}))} \tag{3}$$

where the score function of the local-local objective is $f_{m,n}(x_t, x_{t+1}) = \phi_{m,n}(x_t)^T W_l \phi_{m,n}(x_{t+1})$

## 3   The Atari Annotated RAM Interface (AtariARI)

Measuring the usefulness of a representation is still an open problem, as a core utility of representations is their use as feature extractors in tasks that are different from those used for training (e.g., *transfer learning*). Measuring classification performance, for example, may only reveal the amount of class-relevant information in a representation, but may not reveal other information useful for segmentation. It would be useful, then, to have a more *general* set of measures on the usefulness of a representation, such as ones that may indicate more general utility across numerous real-world tasks. In this vein, we assert that in the context of dynamic, visual, interactive environments, the capability of a representation to capture the underlying high-level factors of the state of an environment will be generally useful for a variety of downstream tasks such as prediction, control, and tracking.

We find video games to be a useful candidate for evaluating visual representation learning algorithms primarily because they are spatiotemporal in nature, which is (1) more realistic compared to static i.i.d. datasets and (2) prior work [42, 43] have argued that without temporal structure, recovering the true underlying latent factors is undecidable. Apart from this, video games also provide ready access to the underlying ground truth states, unlike real-world datasets, which we need to evaluate performance of different techniques.

**Annotating Atari RAM:**   ALE does not explicitly expose any ground truth state information. However, ALE does expose the RAM state (128 bytes per timestep) which are used by the game programmer to store important state information such as the location of sprites, the state of the clock, or the current room the agent is in. To extract these variables, we consulted commented disassemblies [44] (or source code) of Atari 2600 games which were made available by Engelhardt [45] and Jentzsch and CPUWIZ [46]. We were able to find and verify important state variables for a total of 22 games. Once this information is acquired, combining it with the ALE interface produces a wrapper that can automatically output a state label for every example frame generated from the game. We make this available with an easy-to-use *gym* wrapper, which returns this information with no change to existing code using *gym* interfaces. Table 1 lists the 22 games along with the categories of variables for each game. We describe the meaning of each category in the next section.

**State variable categories:**   We categorize the state variables of all the games among six major categories: agent localization, small object localization, other localization, score/clock/lives/display,

Table 1: Number of ground truth labels available in the benchmark for each game across each category. Localization is shortened for local. See section 3 for descriptions and examples for each category.

| GAME | AGENT LOCAL. | SMALL OBJECT LOCAL. | OTHER LOCAL. | SCORE/CLOCK LIVES DISPLAY | MISC | OVERALL |
|---|---|---|---|---|---|---|
| ASTEROIDS | 2 | 4 | 30 | 3 | 3 | 41 |
| BERZERK | 2 | 4 | 19 | 4 | 5 | 34 |
| BOWLING | 2 | 2 | 0 | 2 | 10 | 16 |
| BOXING | 2 | 0 | 2 | 3 | 0 | 7 |
| BREAKOUT | 1 | 2 | 0 | 1 | 31 | 35 |
| DEMONATTACK | 1 | 1 | 6 | 1 | 1 | 10 |
| FREEWAY | 1 | 0 | 10 | 1 | 0 | 12 |
| FROSTBITE | 2 | 0 | 9 | 4 | 2 | 17 |
| HERO | 2 | 0 | 0 | 3 | 3 | 8 |
| MONTEZUMAREVENGE | 2 | 0 | 4 | 4 | 5 | 15 |
| MSPACMAN | 2 | 0 | 10 | 2 | 3 | 17 |
| PITFALL | 2 | 0 | 3 | 0 | 0 | 5 |
| PONG | 1 | 2 | 1 | 2 | 0 | 6 |
| PRIVATEEYE | 2 | 0 | 2 | 4 | 2 | 10 |
| QBERT | 3 | 0 | 2 | 0 | 0 | 5 |
| RIVERRAID | 1 | 2 | 0 | 2 | 0 | 5 |
| SEAQUEST | 2 | 1 | 8 | 4 | 3 | 18 |
| SPACEINVADERS | 1 | 1 | 2 | 2 | 1 | 7 |
| TENNIS | 2 | 2 | 2 | 2 | 0 | 8 |
| VENTURE | 2 | 0 | 12 | 3 | 1 | 18 |
| VIDEOPINBALL | 2 | 2 | 0 | 2 | 0 | 6 |
| YARSREVENGE | 2 | 4 | 2 | 0 | 0 | 8 |
| TOTAL | 39 | 27 | 124 | 49 | 70 | 308 |

and miscellaneous. **Agent Loc.** (agent localization) refers to state variables that represent the $x$ or $y$ coordinates on the screen of any sprite controllable by actions. **Small Loc.** (small object localization) variables refer to the $x$ or $y$ screen position of small objects, like balls or missiles. Prominent examples include the ball in Breakout and Pong, and the torpedo in Seaquest. **Other Loc.** (other localization) denotes the $x$ or $y$ location of any other sprites, including enemies or large objects to pick up. For example, the location of ghosts in Ms. Pacman or the ice floes in Frostbite. **Score/Clock/Lives/Display** refers to variables that track the score of the game, the clock, or the number of remaining lives the agent has, or some other display variable, like the oxygen meter in Seaquest. **Misc.** (Miscellaneous) consists of state variables that are largely specific to a game, and don't fall within one of the above mentioned categories. Examples include the existence of each block or pin in Breakout and Bowling, the room number in Montezuma's Revenge, or Ms. Pacman's facing direction.

**Probing:** Evaluating representation learning methods is a challenging open problem. The notion of *disentanglement* [47, 48] has emerged as a way to measure the usefulness of a representation [49, 50]. In this work, we focus only on *explicitness*, i.e the degree to which underlying generative factors can be recovered using a *linear* transformation from the learned representation. This is standard methodology in the self-supervised representation learning literature [15, 25, 51, 16, 26]. Specifically, to evaluate a representation we train linear classifiers predicting each state variable, and we report the mean F1 score.

## 4 Related Work

**Unsupervised representation learning via mutual information objectives:** Recent work in unsupervised representation learning have focused on extracting latent representations by maximizing a lower bound on the mutual information between the representation and the input. Belghazi et al. [24] estimate the mutual information with neural networks using the Donsker-Varadhan representation of the KL divergence [52], while Chen et al. [53] use the variational bound from Barber and Agakov [54] to learn discrete latent representations. Hjelm et al. [26] learn representations by maximizing the Jensen-Shannon divergence between joint and product of marginals of an image and its

patches. van den Oord et al. [25] maximize mutual information using a multi-sample version of noise contrastive estimation [37, 41]. See [38] for a review of different variational bounds for mutual information.

**State representation learning:** Learning better state representations is an active area of research within robotics and reinforcement learning. Recently, Cuccu et al. [55] and Eslami et al. [4] show that visual processing and policy learning can be effectively decoupled in pixel-based environments. Jonschkowski and Brock [56] and Jonschkowski et al. [57] propose to learn representations using a set of handcrafted robotic priors. Several prior works use a VAE and its variations to learn a mapping from observations to state representations [50, 18, 58]. Single-view TCN [40] and TDC [59] learn state representations using self-supervised objectives that leverage temporal information in demonstrations. ST-DIM can be considered as an extension of TDC and TCN that also leverages the local spatial structure (see Figure 3b for an ablation of spatial losses in ST-DIM).

A few works have focused on learning state representations that capture factors of an environment that are under the agent's control in order to guide exploration [60, 61] or unsupervised control [62]. [EMI, 61] harnesses mutual information between state embeddings and actions to learn representations that capture just the controllable factors of the environment, like the agent's position. ST-DIM, on the other hand, aims to capture every temporally evolving factor (not just the controllable ones) in an environment, like the position of enemies, score, balls, missiles, moving obstacles, and the agent position. The ST-DIM objective is also different from EMI in that it maximizes the mutual information between global and local representations in consecutive time steps, whereas EMI just considers mutual information between global representations. Lastly, ST-DIM uses an InfoNCE objective instead of the JSD one used in EMI. Our work is also closely related to recent work in learning object-oriented representations [63, 64, 65].

**Evaluation frameworks of representations:** Evaluating representations is an open problem, and doing so is usually domain specific. In vision tasks, it is common to evaluate based on the presence of linearly separable label-relevant information, either in the domain the representation was learned on [66] or in transfer learning tasks [67, 68]. In NLP, the SentEval [69] and GLUE [70] benchmarks provide a means of providing a more linguistic-specific understanding of what the model has learned, and these have become a standard tool in NLP research. Such et al. [71] has shown initial quantitative and qualitative comparisons between the performance and representations of several DRL algorithms. Our evaluation framework can be thought of as a GLUE-like benchmarking tool for RL, providing a fine-grained understanding of how well the RL agent perceives the objects in the scene. Analogous to GLUE in NLP, we anticipate that our benchmarking tool will be useful in RL research in order to better design components of agent learning.

## 5 Experimental Setup

We evaluate the performance of different representation learning methods on our benchmark. Our experimental pipeline consists of first training an encoder, then freezing its weights and evaluating its performance on linear probing tasks. For each identified generative factor in each game, we construct a linear probing task where the representation is trained to predict the ground truth value of that factor. Note that the gradients are not backpropagated through the encoder network, and only used to train the linear classifier on top of the representation.

### 5.1 Data preprocessing and acquisition

We consider two different modes for collecting the data: (1) using a random agent (steps through the environment by selecting actions randomly), and (2) using a PPO [72] agent trained for 50M timesteps. For both these modes, we ensure there is enough data diversity by collecting data using 8 differently initialized workers. We also add additional stochasticity to the pretrained PPO agent by using an $\epsilon$-greedy like mechanism wherein at each timestep we take a random action with probability $\epsilon$ [2].

## 5.2 Methods

In our evaluations, we compare the following methods:

1. Randomly-initialized CNN encoder (RANDOM-CNN).
2. Variational autoencoder (VAE) [12] on raw observations.
3. Next-step pixel prediction model (PIXEL-PRED) inspired by the "No-action Feedforward" model from [22].
4. Contrastive Predictive Coding (CPC) [25], which maximizes the mutual information between current latents and latents at a future timestep.
5. SUPERVISED model which learns the encoder and the linear probe using the labels. The gradients are backpropagated through the encoder in this case, so this provides a best-case performance bound.

All methods use the same base encoder architecture, which is the CNN from [73], but adapted for the full 160x210 Atari frame size. To ensure a fair comparison, we use a representation size of 256 for each method. As a sanity check, we include a blind majority classifier (MAJ-CLF), which predicts label values based on the mode of the train set. More details in Appendix, section A.

## 5.3 Probing

We train a different 256-way[3] linear classifier with the representation under consideration as input. We ensure the distribution of realizations of each state variable has high entropy by pruning any variable with entropy less than 0.6. We also ensure there are no duplicates between the train and test set. We train each linear probe with 35,000 frames and use 5,000 and 10,000 frames each for validation and test respectively. We use early stopping and a learning rate scheduler based on plateaus in the validation loss.

## 6 Results

Table 2: Probe F1 scores averaged across categories for each game (data collected by random agents)

| GAME | MAJ-CLF | RANDOM-CNN | VAE | PIXEL-PRED | CPC | ST-DIM | SUPERVISED |
|---|---|---|---|---|---|---|---|
| ASTEROIDS | 0.28 | 0.34 | 0.36 | 0.34 | 0.42 | **0.49** | 0.52 |
| BERZERK | 0.18 | 0.43 | 0.45 | **0.55** | **0.56** | 0.53 | 0.68 |
| BOWLING | 0.33 | 0.48 | 0.50 | 0.81 | 0.90 | **0.96** | 0.95 |
| BOXING | 0.01 | 0.19 | 0.20 | 0.44 | 0.29 | **0.58** | 0.83 |
| BREAKOUT | 0.17 | 0.51 | 0.57 | 0.70 | 0.74 | **0.88** | 0.94 |
| DEMONATTACK | 0.16 | 0.26 | 0.26 | 0.32 | 0.57 | **0.69** | 0.83 |
| FREEWAY | 0.01 | 0.50 | 0.01 | **0.81** | 0.47 | **0.81** | 0.98 |
| FROSTBITE | 0.08 | 0.57 | 0.51 | 0.72 | **0.76** | 0.75 | 0.85 |
| HERO | 0.22 | 0.75 | 0.69 | 0.74 | 0.90 | **0.93** | 0.98 |
| MONTEZUMAREVENGE | 0.08 | 0.68 | 0.38 | 0.74 | 0.75 | **0.78** | 0.87 |
| MSPACMAN | 0.10 | 0.49 | 0.56 | **0.74** | 0.65 | 0.72 | 0.87 |
| PITFALL | 0.07 | 0.34 | 0.35 | 0.44 | 0.46 | **0.60** | 0.83 |
| PONG | 0.10 | 0.17 | 0.09 | 0.70 | 0.71 | **0.81** | 0.87 |
| PRIVATEEYE | 0.23 | 0.70 | 0.71 | 0.83 | 0.81 | **0.91** | 0.97 |
| QBERT | 0.29 | 0.49 | 0.49 | 0.52 | 0.65 | **0.73** | 0.76 |
| RIVERRAID | 0.04 | 0.34 | 0.26 | **0.41** | **0.40** | 0.36 | 0.57 |
| SEAQUEST | 0.29 | 0.57 | 0.56 | 0.62 | **0.66** | **0.67** | 0.85 |
| SPACEINVADERS | 0.14 | 0.41 | 0.52 | **0.57** | 0.54 | **0.57** | 0.75 |
| TENNIS | 0.09 | 0.41 | 0.29 | 0.57 | **0.60** | **0.60** | 0.81 |
| VENTURE | 0.09 | 0.36 | 0.38 | 0.46 | 0.51 | **0.58** | 0.68 |
| VIDEOPINBALL | 0.09 | 0.37 | 0.45 | 0.57 | 0.58 | **0.61** | 0.82 |
| YARSREVENGE | 0.01 | 0.22 | 0.08 | 0.19 | 0.39 | **0.42** | 0.74 |
| MEAN | 0.14 | 0.44 | 0.40 | 0.58 | 0.61 | **0.68** | 0.82 |

Table 3: Probe F1 scores for different methods averaged across all games for each category (data collected by random agents)

| | | RANDOM | | | | | |
|---|---|---|---|---|---|---|---|
| CATEGORY | MAJ-CLF | CNN | VAE | PIXEL-PRED | CPC | ST-DIM | SUPERVISED |
| SMALL LOC. | 0.14 | 0.19 | 0.18 | 0.31 | 0.42 | **0.51** | 0.66 |
| AGENT LOC. | 0.12 | 0.31 | 0.32 | 0.48 | 0.43 | **0.58** | 0.81 |
| OTHER LOC. | 0.14 | 0.50 | 0.39 | 0.61 | 0.66 | **0.69** | 0.80 |
| SCORE/CLOCK/LIVES/DISPLAY | 0.13 | 0.58 | 0.54 | 0.76 | 0.83 | **0.87** | 0.91 |
| MISC. | 0.26 | 0.59 | 0.63 | 0.70 | 0.71 | **0.75** | 0.83 |

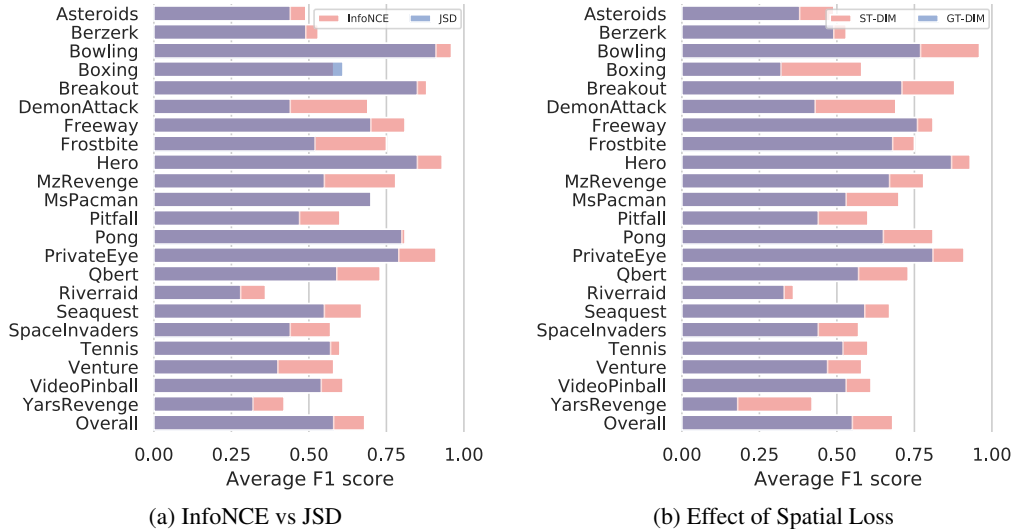

(a) InfoNCE vs JSD

(b) Effect of Spatial Loss

Figure 3: Different ablations for the ST-DIM model

We report the F1 averaged across all categories for each method and for each game in Table 2 for data collected by random agent. In addition, we provide a breakdown of probe results in each category, such as small object localization or score/lives classification in Table 3 for the random agent. We include the corresponding tables for these results with data collected by a pretrained PPO agent in tables 6 and 7. The results in table 2 show that ST-DIM largely outperforms other methods in terms of mean F1 score. In general, contrastive methods (ST-DIM and CPC) methods seem to perform better than generative methods (VAE and PIXEL-PRED) at these probing tasks. We find that RandomCNN is a strong prior in Atari games as has been observed before [74], possibly due to the inductive bias captured by the CNN architecture empirically observed in [75]. We find similar trends to hold on results with data collected by a PPO agent. Despite contrastive methods performing well, there is still a sizable gap between ST-DIM and the fully supervised approach, leaving room for improvement from new unsupervised representation learning techniques for the benchmark.

# 7   Discussion

**Ablations:**   We investigate two ablations of our ST-DIM model: Global-T-DIM, which only maximizes the mutual information between the global representations (similar in construction to PCL [76]) and JSD-ST-DIM, which uses the NCE loss [77] instead of the InfoNCE loss, which is equivalent to maximizing the Jensen Shannon Divergence between representations. We report results from these ablations in Figure 3. We see from the results in that 1) the InfoNCE loss performs better than the JSD loss and 2) contrasting spatiotemporally (and not just temporally) is important across the board for capturing all categories of latent factors.

We found ST-DIM has two main advantages which explain its superior performance over other methods and over its own ablations. It captures small objects much better than other methods, and is more robust to the presence of easy-to-exploit features which hurts other contrastive methods. Both these advantages are due to ST-DIM maximizing mutual information of patch representations.

**Capturing small objects:** As we can see in Table 3, ST-DIM performs better at capturing small objects than other methods, especially generative models like VAE and pixel prediction methods. This is likely because generative models try to model every pixel, so they are not penalized much if they fail to model the few pixels that make up a small object. Similarly, ST-DIM holds this same advantage over Global-T-DIM (see Table 9), which is likely due to the fact that Global-T-DIM is not penalized if its global representation fails to capture features from some patches of the frame.

**Robust to presence of easy-to-exploit features:** Representation learning with mutual information or contrastive losses often fail to capture all salient features if a few easy-to-learn features are sufficient to saturate the objective. This phenomenon has been linked to the looseness of mutual information lower bounds [34, 35] and *gradient starvation* [78]. We see the most prominent example of this phenomenon in Boxing. The observations in Boxing have a clock showing the time remaining in the round. A representation which encodes the shown time can perform near-perfect predictions without learning any other salient features in the observation. Table 4 shows that CPC, Global T-DIM, and ST-DIM perform well at predicting the clock variable. However only ST-DIM does well on encoding the other variables such as the score and the position of the boxers.

We also observe that the best generative model (PIXEL-PRED) does not suffer from this problem. It performs its worst on high-entropy features such as the clock and player score (where ST-DIM excels), and does slightly better than ST-DIM on low-entropy features which have a large contribution in the pixel space such as player and enemy locations. This sheds light on the qualitative difference between contrastive and generative methods: contrastive methods prefer capturing high-entropy features (irrespective of contribution to pixel space) while generative methods do not, and generative methods prefer capturing large objects which have low entropy. This complementary nature suggests hybrid models as an exciting direction of future work.

Table 4: Breakdown of F1 Scores for every state variable in Boxing for ST-DIM, CPC, and Global-T-DIM, an ablation of ST-DIM that removes the spatial contrastive constraint, for the game Boxing

| METHOD | VAE | PIXEL-PRED | CPC | GLOBAL-T-DIM | ST-DIM |
|---|---|---|---|---|---|
| CLOCK | 0.03 | 0.27 | 0.79 | 0.81 | **0.92** |
| ENEMY_SCORE | 0.19 | 0.58 | 0.59 | **0.74** | 0.70 |
| ENEMY_X | 0.32 | 0.49 | 0.15 | 0.17 | **0.51** |
| ENEMY_Y | 0.22 | **0.42** | 0.04 | 0.16 | 0.38 |
| PLAYER_SCORE | 0.08 | 0.32 | 0.56 | 0.45 | **0.88** |
| PLAYER_X | 0.33 | 0.54 | 0.19 | 0.13 | **0.56** |
| PLAYER_Y | 0.16 | **0.43** | 0.04 | 0.14 | 0.37 |

# 8 Conclusion

We present a new representation learning technique which maximizes the mutual information of representations across spatial and temporal axes. We also propose a new benchmark for state representation learning based on the Atari 2600 suite of games to emphasize learning multiple generative factors. We demonstrate that the proposed method excels at capturing the underlying latent factors of a state even for small objects or when a large number of objects are present, which prove difficult for generative and other contrastive techniques, respectively. We have shown that our proposed benchmark can be used to study qualitative and quantitative differences between representation learning techniques, and hope that it will encourage more research in the problem of state representation learning.

## Acknowledgements

We are grateful for the collaborative research environment provided by Mila and Microsoft Research. We thank Aaron Courville, Chris Pal, Remi Tachet, Eric Yuan, Chinwei-Huang, Khimya Khetrapal, Tristan Deleu, and Aravind Srinivas for helpful discussions and feedback during the course of this work. We would also like to thank the developers of PyTorch [79] and Weights&Biases.

## Footnotes

[2] For all our experiments, we used $\epsilon = 0.2$.

[3]Each RAM variable is a single byte thus has 256 possible values ranging from 0 to 255.

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
