[Supplementary Material]

# Supplementary Material for Unsupervised State Representation Learning in Atari

## A  Architecture Details

All architectures below use the same encoder architecture as a base, which is the one used in [1] adapted to work for the full 160x210 frame size as shown in figure 1.

1. **Linear Probe**:
   The linear probe is a linear layer of width 256 with a softmax activation and trained with a cross-entropy loss.

2. **Majority Classifier** (maj-clsf):
   The majority classifier is parameterless and just uses the mode of the distribution of classes from the training set for each state variable and guesses that mode for every example on the test set at test time.

3. **Random-CNN**:
   The Random-CNN is the base encoder with randomly initialzied weights and no training

4. **VAE and Pixel-Pred**:
   The VAE and Pixel Prediction model use the base encoder plus each have an extra 256 wide fully connected layer to parameterize the log variance for the VAE and to more closely resemble the *No Action Feed Forward* model from [2]. In addition bith models have a deconvolutional network as a decoder, which is the exact transpose of the base encoder in figure 1.

5. **CPC**:
   CPC uses the same architecture as described in [3] with our base encoder from figure 1 being used as the image encoder $g_{enc}$.

6. **ST-DIM (and its ablations)**:
   ST-DIM and the two ablations, JSD-ST-DIM and Global-T-DIM, all use the same architecture which is the base encoder plus a 1x256x256 bilinear layer.

7. **Supervised**:
   The supervised model is our base encoder plus our linear probe trained end-to-end with the ground truth labels.

8. **PPO Features** (section E):
   The PPO model is our base encoder plus two linear layers for the policy and the value function, respectively.

## B  Preprocessing and Hyperparameters

We preprocess frames primarily in the same way as described in [1], with the key difference being we use the full 210x160 images for all our experiments instead of downsampling to 84x84. Table 1 lists the hyper-parameters we use across all games. For all our experiments, we use a learning rate scheduler based on plateaus in the validation loss (for both contrastive training and probing).

**Compute infrastructure:**  We run our experiments on a autoscaling-cluster with multiple P100 and V100 GPUs. We use 8 cores per machines to distribute data collection across different workers.

Figure 1: The base encoder architecture used for all models in this work

## C   Results with Probes Trained on Data Collected By a Pretrained RL agent

In addition to evaluating on data collected by a random agent, we also evaluate different representation learning methods on data collected by a pretrained PPO [4] agent. Specifically, we use a PPO agent trained for 50M steps on each game. We choose actions stochastically by sampling from the PPO agent's action distribution at every time step, and inject additional stochasticity by using an $\epsilon$-greedy mechanism with $\epsilon = 0.2$. Table 2 shows the game-by-game breakdown of mean F1 probe scores obtained by each method in this evaluation setting. Table 3 additionally shows the category-wise breakdown of results for each method. We observe a similar trend in performance as observed earlier with a random agent.

## D   More Detailed Ablation Results

We expand on the results reported on different ablations (JSD-ST-DIM and Global-T-DIM) of STDIM in the main text, and provide a game by game breakdown of results in Table 4, and a category-wise breakdown in Table 5. We also include an additional Static-DIM ablation which gets rid of any temporal context in the contrastive task by sampling negatives from a different game.

Table 1: Preprocessing steps and hyperparameters

| Parameter | Value |
|---|---|
| Image Width | 160 |
| Image Height | 210 |
| Grayscaling | Yes |
| Action Repetitions | 4 |
| Max-pool over last N action repeat frames | 2 |
| Frame Stacking | None |
| End of episode when life lost | Yes |
| No-Op action reset | Yes |
| Batch size | 64 |
| Sequence Length (CPC) | 100 |
| Learning Rate (Training) | 3e-4 |
| Learning Rate (Probing) | 3e-4 |
| Entropy Threshold | 0.6 |
| Encoder training steps | 70000 |
| Probe training steps | 35000 |
| Probe test steps | 10000 |

Table 2: Probe F1 scores for all games for data collected by a pretrained PPO (50M steps) agent

| GAME | MEAN AGENT REWARDS | MAJ-CLF | RANDOM-CNN | VAE | PIXEL-PRED | CPC | ST-DIM | SUPERVISED |
|---|---|---|---|---|---|---|---|---|
| ASTEROIDS | 489862.00 | 0.23 | 0.31 | 0.35 | 0.31 | 0.38 | **0.40** | 0.56 |
| BERZERK | 1913.00 | 0.13 | 0.33 | 0.35 | 0.39 | 0.38 | **0.43** | 0.61 |
| BOWLING | 29.80 | 0.23 | 0.61 | 0.51 | 0.81 | 0.90 | **0.98** | 0.98 |
| BOXING | 93.30 | 0.05 | 0.30 | 0.32 | 0.57 | 0.32 | **0.66** | 0.87 |
| BREAKOUT | 580.40 | 0.09 | 0.34 | 0.59 | 0.47 | 0.55 | **0.66** | 0.87 |
| DEMONATTACK | 428165.00 | 0.03 | 0.19 | 0.18 | 0.26 | 0.43 | **0.58** | 0.76 |
| FREEWAY | 33.50 | 0.01 | 0.36 | 0.02 | **0.60** | 0.38 | **0.60** | 0.76 |
| FROSTBITE | 3561.00 | 0.13 | 0.57 | 0.46 | 0.70 | **0.74** | 0.69 | 0.85 |
| HERO | 44999.00 | 0.12 | 0.54 | 0.60 | 0.68 | **0.86** | 0.77 | 0.96 |
| MZREVENGE | 0.00 | 0.08 | 0.68 | 0.58 | 0.72 | **0.77** | **0.76** | 0.88 |
| MSPACMAN | 4588.00 | 0.07 | 0.34 | 0.36 | **0.52** | 0.45 | 0.49 | 0.71 |
| PITFALL | 0.00 | 0.16 | 0.39 | 0.37 | 0.53 | 0.69 | **0.74** | 0.92 |
| PONG | 21.00 | 0.02 | 0.10 | 0.24 | 0.67 | 0.63 | **0.79** | 0.87 |
| PRIVATEEYE | -10.00 | 0.24 | 0.71 | 0.69 | 0.87 | 0.83 | **0.91** | 0.99 |
| QBERT | 30590.00 | 0.06 | 0.36 | 0.38 | 0.39 | **0.51** | 0.48 | 0.65 |
| RIVERRAID | 20632.00 | 0.04 | 0.25 | 0.21 | **0.34** | 0.31 | 0.22 | 0.59 |
| SEAQUEST | 1620.00 | 0.29 | 0.64 | 0.58 | **0.75** | 0.69 | **0.75** | 0.90 |
| SPACEINVADERS | 2892.50 | 0.02 | 0.28 | 0.30 | **0.41** | 0.32 | **0.41** | 0.65 |
| TENNIS | -4.30 | 0.15 | 0.25 | 0.13 | **0.65** | 0.63 | **0.65** | 0.61 |
| VENTURE | 0.00 | 0.05 | 0.32 | 0.36 | 0.37 | 0.50 | **0.59** | 0.69 |
| VIDEOPINBALL | 356362.00 | 0.13 | 0.36 | 0.42 | **0.56** | **0.57** | 0.54 | 0.79 |
| YARSREVENGE | 5520.00 | 0.03 | 0.14 | 0.26 | 0.23 | 0.38 | **0.43** | 0.74 |
| MEAN | - | 0.11 | 0.38 | 0.38 | 0.54 | 0.56 | **0.62** | 0.78 |

# E   Probing Pretrained RL Agents

We make a first attempt at examining the features that RL agents learn. Specifically, we train linear probes on the representations from PPO agents that were trained for 50 million frames. The architecture of the PPO agent is described in section A. As we see from table 6, the features perform poorly in the probing tasks compared to the baselines. Kansky et al. [5], Zhang et al. [6] have also argued that model-free agents have trouble encoding high level state information. However, we note that these are preliminary results and require thorough investigation over different policies and models.

Table 3: Probe F1 scores for different methods averaged across all games for each category (data collected by a pretrained PPO (50M steps) agent

| CATEGORY | MAJ-CLF | RANDOM-CNN | VAE | PIXEL-PRED | CPC | ST-DIM | SUPERVISED |
|---|---|---|---|---|---|---|---|
| SMALL LOC. | 0.10 | 0.13 | 0.14 | 0.27 | 0.31 | **0.41** | 0.65 |
| AGENT LOC. | 0.11 | 0.34 | 0.34 | 0.48 | 0.45 | **0.54** | 0.83 |
| OTHER LOC. | 0.14 | 0.47 | 0.38 | 0.56 | 0.58 | **0.61** | 0.74 |
| SCORE/CLOCK/LIVES/DISPLAY | 0.05 | 0.44 | 0.50 | 0.71 | 0.74 | **0.80** | 0.90 |
| MISC. | 0.19 | 0.53 | 0.57 | 0.62 | 0.65 | **0.67** | 0.83 |

Table 4: Probe F1 scores for different ablations of ST-DIM for all games averaged across each category (data collected by random agents)

| GAME | STATIC-DIM | JSD-ST-DIM | GLOBAL-T-DIM | ST-DIM |
|---|---|---|---|---|
| ASTEROIDS | 0.37 | 0.44 | 0.38 | **0.49** |
| BERZERK | 0.41 | 0.49 | 0.49 | **0.53** |
| BOWLING | 0.34 | 0.91 | 0.77 | **0.96** |
| BOXING | 0.09 | **0.61** | 0.32 | 0.58 |
| BREAKOUT | 0.19 | 0.85 | 0.71 | **0.88** |
| DEMONATTACK | 0.30 | 0.44 | 0.43 | **0.69** |
| FREEWAY | 0.02 | 0.70 | 0.76 | **0.81** |
| FROSTBITE | 0.27 | 0.52 | 0.68 | **0.75** |
| HERO | 0.59 | 0.85 | 0.87 | **0.93** |
| MONTEZUMAREVENGE | 0.17 | 0.55 | 0.67 | **0.78** |
| MSPACMAN | 0.17 | 0.70 | 0.53 | **0.72** |
| PITFALL | 0.22 | 0.47 | 0.44 | **0.60** |
| PONG | 0.13 | **0.80** | 0.65 | **0.81** |
| PRIVATEEYE | 0.25 | 0.79 | 0.81 | **0.91** |
| QBERT | 0.41 | 0.59 | 0.57 | **0.73** |
| RIVERRAID | 0.16 | 0.28 | 0.33 | **0.36** |
| SEAQUEST | 0.41 | 0.55 | 0.59 | **0.67** |
| SPACEINVADERS | 0.40 | 0.44 | 0.44 | **0.57** |
| TENNIS | 0.17 | 0.57 | 0.52 | **0.60** |
| VENTURE | 0.25 | 0.40 | 0.47 | **0.58** |
| VIDEOPINBALL | 0.21 | 0.54 | 0.53 | **0.61** |
| YARSREVENGE | 0.12 | 0.32 | 0.18 | **0.42** |
| MEAN | 0.26 | 0.58 | 0.55 | **0.68** |

# F  Accuracy Metric

In tables 7 and 8, we report the game by game and categorical probe results for each method, but we use a standard percent accuracy metric instead of the F1 score.

# G  Fine-Grained Results

In tables 9 to 30, we present the F1 scores for every game, for every method, for every state variable for data collected by a random agent. We further confirm that ST-DIM is very good at capturing small objects compared to generative models, specifically in Breakout and Video Pinball (tables 13 and 29). We also see that CPC is very good at capturing objects that move very regularly and predictably, like ice flows in Frostbite as in table 16 and divers in Seaquest as in table 25. We also present in tables 31 to 52 fine grained results for data collected by a pretrained PPO agent.

Table 5: Different ablations of ST-DIM. F1 scores for for each category averaged across all games (data collected by random agents)

|  | STATIC-DIM | JSD-ST-DIM | GLOBAL-T-DIM | ST-DIM |
|---|---|---|---|---|
| SMALL LOC. | 0.18 | 0.44 | 0.37 | **0.51** |
| AGENT LOC. | 0.19 | 0.47 | 0.43 | **0.58** |
| OTHER LOC. | 0.27 | 0.64 | 0.53 | **0.69** |
| SCORE/CLOCK/LIVES/DISPLAY | 0.33 | 0.69 | 0.76 | **0.87** |
| MISC. | 0.41 | 0.64 | 0.66 | **0.75** |

Table 6: Probe results on features from a PPO agent trained on 50 million timesteps compared with a majority classifier and random-cnn baseline. The probes for all three methods are trained with data from the PPO agent that was trained for 50M frames

|  | MAJ-CLF | RANDOM-CNN | PRETRAINED-RL-AGENT |
|---|---|---|---|
| ASTEROIDS | 0.23 | **0.31** | **0.31** |
| BERZERK | 0.13 | **0.33** | 0.30 |
| BOWLING | 0.23 | **0.61** | 0.48 |
| BOXING | 0.05 | **0.30** | 0.12 |
| BREAKOUT | 0.09 | **0.34** | 0.23 |
| DEMONATTACK | 0.03 | **0.19** | 0.16 |
| FREEWAY | 0.01 | **0.36** | 0.26 |
| FROSTBITE | 0.13 | **0.57** | 0.43 |
| HERO | 0.12 | **0.54** | 0.42 |
| MONTEZUMAREVENGE | 0.08 | **0.68** | 0.07 |
| MSPACMAN | 0.06 | **0.34** | 0.26 |
| PITFALL | 0.16 | **0.39** | 0.23 |
| PONG | 0.02 | **0.10** | 0.09 |
| PRIVATEEYE | 0.24 | **0.71** | 0.31 |
| QBERT | 0.06 | **0.36** | 0.34 |
| RIVERRAID | 0.04 | **0.25** | 0.10 |
| SEAQUEST | 0.29 | **0.64** | 0.50 |
| SPACEINVADERS | 0.02 | **0.28** | 0.19 |
| TENNIS | 0.15 | 0.25 | **0.66** |
| VENTURE | 0.05 | **0.32** | 0.08 |
| VIDEOPINBALL | 0.13 | **0.36** | 0.21 |
| YARSREVENGE | 0.03 | **0.14** | 0.09 |
| MEAN | 0.11 | **0.38** | 0.27 |

Table 7: Probe Accuracy scores averaged across categories for each game (data collected by random agents)

| GAME | MAJ-CLF | RANDOM-CNN | VAE | PIXEL-PRED | CPC | ST-DIM | SUPERVISED |
|---|---|---|---|---|---|---|---|
| ASTEROIDS | 0.37 | 0.42 | 0.41 | 0.43 | 0.48 | **0.52** | 0.53 |
| BERZERK | 0.30 | 0.48 | 0.46 | **0.56** | **0.57** | 0.54 | 0.69 |
| BOWLING | 0.43 | 0.54 | 0.56 | 0.83 | 0.90 | **0.96** | 0.95 |
| BOXING | 0.05 | 0.22 | 0.23 | 0.45 | 0.32 | **0.59** | 0.83 |
| BREAKOUT | 0.28 | 0.55 | 0.61 | 0.71 | 0.75 | **0.89** | 0.94 |
| DEMONATTACK | 0.26 | 0.30 | 0.31 | 0.35 | 0.58 | **0.70** | 0.83 |
| FREEWAY | 0.06 | 0.53 | 0.07 | **0.85** | 0.49 | 0.82 | 0.99 |
| FROSTBITE | 0.19 | 0.59 | 0.54 | 0.72 | **0.76** | **0.75** | 0.85 |
| HERO | 0.34 | 0.78 | 0.72 | 0.75 | 0.90 | **0.93** | 0.98 |
| MONTEZUMAREVENGE | 0.16 | 0.70 | 0.41 | 0.74 | 0.76 | **0.78** | 0.87 |
| MSPACMAN | 0.22 | 0.54 | 0.60 | **0.75** | 0.67 | 0.73 | 0.87 |
| PITFALL | 0.20 | 0.42 | 0.35 | 0.47 | 0.49 | **0.61** | 0.83 |
| PONG | 0.20 | 0.26 | 0.19 | 0.72 | 0.73 | **0.82** | 0.88 |
| PRIVATEEYE | 0.35 | 0.72 | 0.72 | 0.83 | 0.81 | **0.91** | 0.97 |
| QBERT | 0.42 | 0.52 | 0.53 | 0.54 | 0.66 | **0.74** | 0.76 |
| RIVERRAID | 0.13 | 0.40 | 0.31 | **0.43** | 0.41 | 0.37 | 0.58 |
| SEAQUEST | 0.43 | 0.63 | 0.61 | 0.65 | **0.69** | **0.69** | 0.85 |
| SPACEINVADERS | 0.24 | 0.46 | 0.57 | **0.61** | 0.57 | 0.59 | 0.76 |
| TENNIS | 0.22 | 0.49 | 0.37 | 0.59 | **0.61** | **0.61** | 0.81 |
| VENTURE | 0.19 | 0.40 | 0.43 | 0.48 | 0.52 | **0.59** | 0.68 |
| VIDEOPINBALL | 0.16 | 0.39 | 0.47 | 0.58 | 0.59 | **0.61** | 0.82 |
| YARSREVENGE | 0.05 | 0.25 | 0.11 | 0.20 | 0.41 | **0.43** | 0.74 |
| MEAN | 0.24 | 0.48 | 0.44 | 0.60 | 0.62 | **0.69** | 0.82 |

Table 8: Probe Accuracy scores for different methods averaged across all games for each category (data collected by random agents)

| CATEGORY | MAJ-CLF | RANDOM CNN | VAE | PIXEL-PRED | CPC | ST-DIM | SUPERVISED |
|---|---|---|---|---|---|---|---|
| SMALL LOC. | 0.23 | 0.29 | 0.26 | 0.36 | 0.46 | **0.53** | 0.67 |
| AGENT LOC. | 0.21 | 0.37 | 0.37 | 0.51 | 0.46 | **0.59** | 0.81 |
| OTHER LOC. | 0.22 | 0.54 | 0.42 | 0.63 | 0.67 | **0.70** | 0.80 |
| SCORE/CLOCK/LIVES/DISPLAY | 0.24 | 0.61 | 0.56 | 0.77 | 0.84 | **0.87** | 0.91 |
| MISC. | 0.38 | 0.61 | 0.65 | 0.71 | 0.72 | **0.75** | 0.83 |

Table 9: Asteroids fine-grained results. Breakdown of F1 Scores for every state variable in Asteroids for every method for probes where data was collected by random agent

| METHOD | MAJ-CLF | RANDOM-CNN | VAE | PIXEL-PRED | CPC | ST-DIM | SUPERVISED |
|---|---|---|---|---|---|---|---|
| ENEMY_ASTEROIDS_X_0 | 0.02 | 0.33 | **0.37** | 0.20 | **0.37** | 0.34 | 0.48 |
| ENEMY_ASTEROIDS_X_10 | 0.00 | 0.17 | 0.11 | 0.16 | 0.22 | **0.25** | 0.28 |
| ENEMY_ASTEROIDS_X_11 | 0.03 | 0.11 | 0.07 | 0.14 | 0.20 | **0.26** | 0.22 |
| ENEMY_ASTEROIDS_X_12 | **0.58** | 0.32 | 0.36 | 0.25 | 0.34 | 0.46 | 0.42 |
| ENEMY_ASTEROIDS_X_1 | 0.07 | **0.44** | 0.27 | 0.24 | 0.24 | 0.34 | 0.40 |
| ENEMY_ASTEROIDS_X_2 | 0.15 | **0.41** | 0.28 | 0.16 | 0.39 | 0.34 | 0.41 |
| ENEMY_ASTEROIDS_X_3 | 0.48 | 0.49 | **0.59** | 0.53 | 0.57 | 0.48 | 0.50 |
| ENEMY_ASTEROIDS_X_7 | 0.00 | 0.34 | 0.25 | 0.27 | **0.39** | 0.36 | 0.66 |
| ENEMY_ASTEROIDS_X_8 | 0.00 | 0.24 | 0.13 | 0.18 | 0.27 | **0.29** | 0.30 |
| ENEMY_ASTEROIDS_X_9 | 0.00 | 0.24 | 0.12 | 0.15 | **0.29** | **0.29** | 0.29 |
| ENEMY_ASTEROIDS_Y_0 | 0.00 | 0.28 | 0.19 | 0.05 | **0.61** | 0.58 | 0.96 |
| ENEMY_ASTEROIDS_Y_10 | **0.65** | 0.50 | 0.60 | 0.53 | 0.61 | **0.66** | 0.83 |
| ENEMY_ASTEROIDS_Y_12 | 0.13 | 0.84 | 0.62 | 0.84 | 0.79 | **0.86** | 0.87 |
| ENEMY_ASTEROIDS_Y_1 | 0.07 | 0.31 | 0.18 | 0.08 | 0.41 | **0.46** | 0.83 |
| ENEMY_ASTEROIDS_Y_2 | 0.15 | 0.24 | 0.23 | 0.15 | 0.38 | **0.45** | 0.69 |
| ENEMY_ASTEROIDS_Y_3 | 0.48 | 0.49 | **0.60** | 0.53 | **0.61** | **0.61** | 0.71 |
| ENEMY_ASTEROIDS_Y_5 | 0.68 | 0.76 | 0.83 | 0.64 | 0.79 | **0.89** | 0.75 |
| ENEMY_ASTEROIDS_Y_7 | 0.00 | 0.25 | 0.18 | 0.03 | 0.57 | **0.67** | 0.94 |
| ENEMY_ASTEROIDS_Y_8 | 0.04 | 0.25 | 0.16 | 0.04 | 0.45 | **0.53** | 0.78 |
| ENEMY_ASTEROIDS_Y_9 | 0.21 | 0.13 | 0.23 | 0.09 | 0.42 | **0.52** | 0.72 |
| NUM_LIVES_DIRECTION | 0.01 | 0.05 | 0.10 | 0.07 | 0.14 | **0.19** | 0.12 |
| PLAYER_MISSILE1_DIRECTION | 0.79 | 0.79 | 0.81 | 0.81 | **0.83** | **0.83** | 0.82 |
| PLAYER_MISSILE2_DIRECTION | 0.60 | 0.62 | 0.62 | 0.60 | 0.62 | **0.64** | 0.68 |
| PLAYER_MISSILE_X1 | 0.02 | 0.04 | 0.04 | 0.06 | 0.08 | **0.12** | 0.07 |
| PLAYER_MISSILE_X2 | 0.02 | 0.04 | 0.04 | 0.06 | 0.07 | **0.11** | 0.08 |
| PLAYER_MISSILE_Y1 | 0.78 | 0.79 | 0.80 | 0.81 | **0.83** | **0.83** | 0.83 |
| PLAYER_MISSILE_Y2 | 0.60 | 0.61 | 0.61 | 0.59 | 0.61 | **0.63** | 0.74 |
| PLAYER_SCORE_LOW | 0.02 | 0.24 | 0.45 | 0.34 | 0.47 | **0.92** | 0.94 |
| PLAYER_X | 0.03 | 0.05 | 0.05 | 0.05 | 0.08 | **0.12** | 0.11 |
| PLAYER_Y | 0.61 | 0.62 | 0.60 | 0.60 | 0.65 | **0.68** | 0.86 |

Table 10: Berzerk fine-grained results. Breakdown of F1 Scores for every state variable in Berzerk for every method for probes where data was collected by random agent

| METHOD | MAJ-CLF | RANDOM-CNN | VAE | PIXEL-PRED | CPC | ST-DIM | SUPERVISED |
|---|---|---|---|---|---|---|---|
| ENEMY_EVILOTTO_Y | 0.09 | 0.73 | 0.71 | 0.79 | **0.84** | 0.82 | 0.89 |
| ENEMY_ROBOTS_X_0 | 0.04 | 0.35 | 0.46 | **0.54** | 0.52 | 0.43 | 0.86 |
| ENEMY_ROBOTS_X_1 | 0.04 | 0.23 | 0.37 | 0.44 | **0.50** | 0.45 | 0.68 |
| ENEMY_ROBOTS_X_2 | 0.07 | 0.34 | 0.33 | 0.48 | **0.50** | 0.44 | 0.53 |
| ENEMY_ROBOTS_X_3 | 0.18 | 0.28 | 0.36 | 0.47 | **0.49** | **0.49** | 0.57 |
| ENEMY_ROBOTS_X_4 | 0.55 | 0.46 | 0.47 | **0.66** | 0.61 | 0.59 | 0.69 |
| ENEMY_ROBOTS_X_5 | **0.86** | 0.72 | 0.76 | 0.76 | 0.77 | 0.82 | 0.87 |
| ENEMY_ROBOTS_Y_0 | 0.47 | 0.81 | 0.81 | **0.90** | 0.85 | 0.82 | 0.97 |
| ENEMY_ROBOTS_Y_1 | 0.26 | 0.58 | 0.60 | **0.76** | 0.74 | **0.75** | 0.83 |
| ENEMY_ROBOTS_Y_2 | 0.21 | 0.47 | 0.47 | **0.66** | 0.65 | 0.63 | 0.77 |
| ENEMY_ROBOTS_Y_3 | 0.18 | 0.43 | 0.39 | 0.61 | **0.66** | 0.61 | 0.78 |
| ENEMY_ROBOTS_Y_4 | 0.55 | 0.56 | 0.51 | **0.73** | 0.67 | **0.72** | 0.85 |
| GAME_LEVEL | 0.21 | 0.73 | 0.71 | 0.79 | 0.82 | **0.83** | 0.84 |
| NUM_LIVES | 0.21 | 0.73 | 0.71 | 0.77 | 0.82 | **0.83** | 0.86 |
| PLAYER_DIRECTION | 0.03 | 0.15 | **0.17** | **0.18** | 0.15 | 0.16 | 0.35 |
| PLAYER_MISSILE_DIRECTION | 0.07 | 0.21 | 0.18 | 0.28 | 0.29 | **0.30** | 0.56 |
| PLAYER_MISSILE_X | 0.01 | 0.04 | 0.07 | 0.07 | 0.10 | **0.13** | 0.36 |
| PLAYER_MISSILE_Y | 0.00 | 0.04 | 0.04 | 0.08 | 0.10 | **0.12** | 0.39 |
| PLAYER_SCORE_1 | 0.29 | 0.87 | 0.86 | 0.82 | **0.95** | 0.86 | 0.92 |
| PLAYER_SCORE_2 | 0.26 | 0.94 | 0.95 | **0.97** | 0.93 | 0.90 | 0.92 |
| PLAYER_X | 0.02 | 0.22 | 0.33 | **0.61** | 0.50 | 0.46 | 0.78 |
| PLAYER_Y | 0.02 | 0.08 | 0.09 | **0.20** | 0.19 | 0.18 | 0.59 |
| ROBOT_MISSILE_DIRECTION | 0.31 | 0.54 | 0.59 | 0.69 | **0.76** | 0.71 | 0.67 |
| ROBOT_MISSILE_X | 0.31 | 0.38 | 0.41 | 0.46 | **0.60** | 0.43 | 0.48 |
| ROBOT_MISSILE_Y | 0.32 | 0.40 | 0.47 | 0.62 | **0.65** | 0.50 | 0.51 |
| ROBOTS_KILLED_COUNT | 0.14 | 0.61 | 0.50 | **0.73** | 0.70 | 0.60 | 0.64 |

Table 11: Bowling fine-grained results. Breakdown of F1 Scores for every state variable in Bowling for every method for probes where data was collected by random agent

| METHOD | MAJ-CLF | RANDOM-CNN | VAE | PIXEL-PRED | CPC | ST-DIM | SUPERVISED |
|---|---|---|---|---|---|---|---|
| BALL_X | 0.00 | 0.01 | 0.01 | 0.31 | 0.60 | **0.81** | 0.80 |
| BALL_Y | 0.44 | 0.34 | 0.41 | 0.74 | 0.83 | **0.96** | 0.99 |
| FRAME_NUMBER_DISPLAY | 0.01 | 0.79 | 0.84 | **1.00** | **1.00** | **1.00** | 1.00 |
| PIN_EXISTENCE_3 | 0.44 | 0.72 | 0.87 | 0.99 | **1.00** | **1.00** | 1.00 |
| PIN_EXISTENCE_6 | 0.42 | 0.86 | 0.78 | 0.94 | **0.98** | 0.99 | 0.99 |
| PIN_EXISTENCE_7 | 0.44 | 0.90 | 0.88 | 0.99 | 0.99 | **1.00** | 1.00 |
| SCORE | 0.01 | 0.09 | 0.08 | 0.89 | **0.97** | 0.89 | 0.94 |

Table 12: Boxing fine-grained results. Breakdown of F1 Scores for every state variable in Boxing for every method for probes where data was collected by random agent

| METHOD | MAJ-CLF | RANDOM-CNN | VAE | PIXEL-PRED | CPC | ST-DIM | SUPERVISED |
|---|---|---|---|---|---|---|---|
| CLOCK | 0.00 | 0.05 | 0.03 | 0.27 | 0.79 | **0.92** | 0.97 |
| ENEMY_SCORE | 0.00 | 0.06 | 0.19 | 0.58 | 0.59 | **0.70** | 1.00 |
| ENEMY_X | 0.00 | 0.37 | 0.32 | 0.49 | 0.15 | **0.51** | 0.80 |
| ENEMY_Y | 0.01 | 0.27 | 0.22 | **0.42** | 0.04 | 0.38 | 0.69 |
| PLAYER_SCORE | 0.00 | 0.03 | 0.08 | 0.32 | 0.56 | **0.88** | 1.00 |
| PLAYER_X | 0.03 | 0.25 | 0.33 | 0.54 | 0.19 | **0.56** | 0.81 |
| PLAYER_Y | 0.00 | 0.18 | 0.16 | **0.43** | 0.04 | 0.37 | 0.72 |

Table 13: Breakout fine-grained results. Breakdown of F1 Scores for every state variable in Breakout for every method for probes where data was collected by random agent

| METHOD | MAJ-CLF | RANDOM-CNN | VAE | PIXEL-PRED | CPC | ST-DIM | SUPERVISED |
|---|---|---|---|---|---|---|---|
| BALL_X | 0.00 | 0.03 | 0.01 | 0.13 | 0.42 | **0.67** | 0.77 |
| BALL_Y | 0.00 | 0.16 | 0.01 | 0.16 | 0.59 | **0.72** | 0.85 |
| BLOCKS_HIT_COUNT | 0.19 | 0.94 | **0.96** | **0.96** | 0.97 | 0.94 | 1.00 |
| PLAYER_X | 0.05 | 0.05 | 0.33 | 0.70 | 0.51 | **0.91** | 0.97 |
| SCORE | 0.19 | 0.94 | **0.96** | **0.96** | 0.97 | 0.94 | 1.00 |

Table 14: Demonattack fine-grained results. Breakdown of F1 Scores for every state variable in Demonattack for every method for probes where data was collected by random agent

| METHOD | MAJ-CLF | RANDOM-CNN | VAE | PIXEL-PRED | CPC | ST-DIM | SUPERVISED |
|---|---|---|---|---|---|---|---|
| ENEMY_X1 | 0.01 | 0.02 | 0.02 | 0.10 | **0.20** | 0.14 | 0.29 |
| ENEMY_X2 | 0.00 | 0.07 | 0.01 | 0.07 | **0.15** | 0.10 | 0.24 |
| ENEMY_X3 | 0.00 | 0.06 | 0.01 | **0.10** | 0.11 | **0.10** | 0.26 |
| ENEMY_Y1 | 0.01 | 0.11 | 0.02 | 0.14 | **0.21** | 0.15 | 0.34 |
| ENEMY_Y2 | 0.01 | 0.08 | 0.02 | 0.10 | **0.17** | 0.14 | 0.31 |
| ENEMY_Y3 | 0.01 | 0.13 | 0.03 | 0.14 | **0.19** | **0.19** | 0.40 |
| LEVEL | 0.59 | 0.77 | 0.81 | 0.90 | **0.99** | 0.97 | 0.99 |
| MISSILE_Y | 0.03 | 0.03 | 0.03 | 0.13 | 0.46 | **0.84** | 0.97 |
| NUM_LIVES | 0.19 | 0.37 | 0.42 | 0.40 | **1.00** | 0.98 | 1.00 |
| PLAYER_X | 0.00 | 0.04 | 0.01 | 0.07 | 0.23 | **0.54** | 0.89 |

Table 15: Freeway fine-grained results. Breakdown of F1 Scores for every state variable in Freeway for every method for probes where data was collected by random agent

| METHOD | MAJ-CLF | RANDOM-CNN | VAE | PIXEL-PRED | CPC | ST-DIM | SUPERVISED |
|---|---|---|---|---|---|---|---|
| ENEMY_CAR_X_0 | 0.00 | 0.92 | 0.00 | **1.00** | **0.99** | **1.00** | 1.00 |
| ENEMY_CAR_X_1 | 0.00 | 0.97 | 0.00 | 0.99 | **0.99** | **1.00** | 1.00 |
| ENEMY_CAR_X_2 | 0.00 | 0.96 | 0.00 | **1.00** | 0.96 | **1.00** | 1.00 |
| ENEMY_CAR_X_3 | 0.00 | 0.94 | 0.00 | **0.99** | 0.86 | **1.00** | 1.00 |
| ENEMY_CAR_X_4 | 0.00 | 0.85 | 0.00 | 0.98 | 0.56 | **1.00** | 1.00 |
| ENEMY_CAR_X_5 | 0.00 | 0.85 | 0.00 | 0.98 | 0.57 | **1.00** | 1.00 |
| ENEMY_CAR_X_6 | 0.00 | 0.97 | 0.00 | **0.99** | 0.80 | **1.00** | 1.00 |
| ENEMY_CAR_X_7 | 0.00 | 0.93 | 0.00 | **0.99** | 0.84 | **1.00** | 1.00 |
| ENEMY_CAR_X_8 | 0.00 | 0.94 | 0.00 | **0.99** | 0.90 | **1.00** | 1.00 |
| ENEMY_CAR_X_9 | 0.00 | 0.85 | 0.00 | **0.99** | **0.99** | **1.00** | 1.00 |
| PLAYER_Y | 0.02 | 0.09 | 0.03 | **0.63** | 0.09 | **0.63** | 0.97 |

Table 16: Frostbite fine-grained results. Breakdown of F1 Scores for every state variable in Frostbite for every method for probes where data was collected by random agent

| METHOD | MAJ-CLF | RANDOM-CNN | VAE | PIXEL-PRED | CPC | ST-DIM | SUPERVISED |
|---|---|---|---|---|---|---|---|
| ENEMY_X_1 | 0.30 | 0.68 | 0.66 | 0.81 | 0.89 | **0.92** | 0.99 |
| ENEMY_X_2 | 0.67 | 0.80 | 0.84 | 0.88 | **0.94** | **0.95** | 1.00 |
| ENEMY_X_3 | 0.17 | 0.75 | 0.63 | 0.78 | **0.89** | **0.90** | 1.00 |
| FOURTH_ROW_ICEFLOW_X | 0.01 | **0.97** | 0.78 | 0.95 | 0.94 | **0.96** | 0.99 |
| IGLOO_BLOCKS_COUNT | 0.04 | 0.66 | 0.65 | 0.93 | **0.97** | **0.97** | 0.99 |
| NUM_LIVES | 0.10 | 0.82 | 0.76 | **1.00** | **1.00** | 0.99 | 1.00 |
| PLAYER_DIRECTION | 0.02 | 0.11 | 0.10 | **0.13** | **0.13** | 0.12 | 0.18 |
| PLAYER_X | 0.03 | 0.16 | 0.14 | 0.29 | **0.42** | 0.41 | 0.72 |
| PLAYER_Y | 0.10 | 0.55 | 0.34 | **0.76** | 0.74 | 0.71 | 0.90 |
| SCORE_2 | 0.04 | 0.51 | 0.63 | 0.84 | **0.95** | 0.87 | 1.00 |
| SECOND_ROW_ICEFLOW_X | 0.01 | **0.94** | 0.80 | **0.94** | **0.94** | **0.95** | 1.00 |
| THIRD_ROW_ICEFLOW_X | 0.01 | **0.96** | 0.78 | **0.96** | 0.93 | **0.96** | 0.99 |
| TOP_ROW_ICEFLOW_X | 0.01 | 0.93 | 0.74 | **0.94** | 0.91 | **0.95** | 0.99 |

Table 17: Hero fine-grained results. Breakdown of F1 Scores for every state variable in Hero for every method for probes where data was collected by random agent

| METHOD | MAJ-CLF | RANDOM-CNN | VAE | PIXEL-PRED | CPC | ST-DIM | SUPERVISED |
|---|---|---|---|---|---|---|---|
| DYNAMITE_COUNT | 0.13 | 0.76 | 0.47 | 0.75 | **0.98** | **0.98** | 1.00 |
| PLAYER_X | 0.00 | 0.59 | 0.56 | 0.61 | **0.79** | **0.80** | 0.99 |
| PLAYER_Y | 0.71 | 0.81 | 0.69 | 0.81 | 0.89 | **0.91** | 0.98 |
| POWER_METER | 0.01 | 0.65 | 0.29 | 0.34 | 0.84 | **0.88** | 0.90 |
| SCORE_0 | 0.37 | 0.86 | 0.92 | 0.95 | 0.89 | **0.98** | 1.00 |
| SCORE_1 | 0.17 | 0.86 | 0.93 | 0.95 | 0.89 | **0.98** | 1.00 |

Table 18: Montezumarevenge fine-grained results. Breakdown of F1 Scores for every state variable in Montezumarevenge for every method for probes where data was collected by random agent

| METHOD | MAJ-CLF | RANDOM-CNN | VAE | PIXEL-PRED | CPC | ST-DIM | SUPERVISED |
|---|---|---|---|---|---|---|---|
| ENEMY_SKULL_X | 0.00 | 0.66 | 0.57 | 0.70 | **0.78** | 0.76 | 0.84 |
| ITEMS_IN_INVENTORY_COUNT | 0.33 | 0.57 | 0.54 | 0.61 | 0.63 | **0.65** | 0.91 |
| NUM_LIVES | 0.02 | **0.99** | 0.11 | 0.95 | **1.00** | **1.00** | 1.00 |
| PLAYER_DIRECTION | 0.03 | 0.49 | 0.40 | 0.58 | **0.62** | 0.60 | 0.71 |
| PLAYER_X | 0.37 | 0.69 | 0.54 | 0.76 | 0.71 | **0.78** | 0.91 |
| PLAYER_Y | 0.01 | 0.66 | 0.48 | 0.81 | 0.74 | **0.85** | 0.94 |
| ROOM_STATE | 0.01 | 0.16 | 0.08 | 0.33 | 0.28 | **0.39** | 0.54 |

Table 19: Mspacman fine-grained results. Breakdown of F1 Scores for every state variable in Mspacman for every method for probes where data was collected by random agent

| METHOD | MAJ-CLF | RANDOM-CNN | VAE | PIXEL-PRED | CPC | ST-DIM | SUPERVISED |
|---|---|---|---|---|---|---|---|
| DOTS_EATEN_COUNT | 0.00 | 0.31 | 0.88 | **0.89** | 0.65 | **0.90** | 0.99 |
| ENEMY_BLINKY_X | 0.03 | 0.27 | 0.26 | **0.59** | **0.60** | 0.54 | 0.70 |
| ENEMY_BLINKY_Y | 0.08 | 0.31 | 0.30 | 0.63 | 0.64 | **0.66** | 0.74 |
| ENEMY_INKY_X | 0.05 | 0.43 | 0.33 | 0.62 | **0.64** | 0.60 | 0.74 |
| ENEMY_INKY_Y | 0.08 | 0.44 | 0.37 | 0.67 | **0.70** | 0.67 | 0.77 |
| ENEMY_PINKY_X | 0.05 | 0.42 | 0.36 | 0.64 | **0.67** | 0.62 | 0.74 |
| ENEMY_PINKY_Y | 0.08 | 0.36 | 0.28 | 0.66 | **0.67** | 0.61 | 0.76 |
| ENEMY_SUE_Y | 0.03 | 0.30 | 0.35 | 0.62 | **0.65** | **0.65** | 0.74 |
| NUM_LIVES | 0.10 | 0.98 | **1.00** | **1.00** | **1.00** | **1.00** | 1.00 |
| PLAYER_DIRECTION | 0.25 | 0.48 | 0.41 | **0.56** | 0.53 | 0.52 | 0.92 |
| PLAYER_SCORE | 0.02 | 0.47 | 0.74 | 0.94 | 0.61 | **0.97** | 1.00 |
| PLAYER_X | 0.03 | 0.41 | 0.32 | **0.52** | 0.48 | 0.43 | 0.73 |
| PLAYER_Y | 0.28 | 0.51 | 0.48 | **0.73** | 0.62 | 0.67 | 0.85 |

Table 20: Pitfall fine-grained results. Breakdown of F1 Scores for every state variable in Pitfall for every method for probes where data was collected by random agent

| METHOD | MAJ-CLF | RANDOM-CNN | VAE | PIXEL-PRED | CPC | ST-DIM | SUPERVISED |
|---|---|---|---|---|---|---|---|
| BOTTOM_OF_ROPE_Y | 0.05 | 0.11 | 0.10 | 0.42 | 0.47 | **0.59** | 0.53 |
| ENEMY_LOGS_X | 0.11 | 0.78 | 0.83 | 0.78 | 0.77 | **0.85** | 0.98 |
| ENEMY_SCORPION_X | 0.19 | **0.91** | 0.86 | 0.88 | 0.86 | 0.86 | 0.99 |
| PLAYER_X | 0.01 | 0.06 | 0.09 | 0.14 | 0.17 | **0.31** | 0.74 |
| PLAYER_Y | 0.05 | 0.12 | 0.13 | 0.22 | 0.25 | **0.57** | 0.91 |

Table 21: Pong fine-grained results. Breakdown of F1 Scores for every state variable in Pong for every method for probes where data was collected by random agent

| METHOD | MAJ-CLF | RANDOM-CNN | VAE | PIXEL-PRED | CPC | ST-DIM | SUPERVISED |
|---|---|---|---|---|---|---|---|
| BALL_X | 0.13 | 0.15 | 0.15 | 0.60 | **0.81** | **0.81** | 0.91 |
| BALL_Y | 0.15 | 0.17 | 0.17 | 0.73 | 0.86 | **0.88** | 0.94 |
| ENEMY_SCORE | 0.01 | 0.41 | 0.00 | **0.99** | **1.00** | 0.99 | 1.00 |
| ENEMY_Y | 0.01 | 0.00 | 0.01 | 0.68 | 0.82 | **0.85** | 0.93 |
| PLAYER_SCORE | 0.51 | 0.59 | 0.36 | **1.00** | **1.00** | **1.00** | 0.97 |
| PLAYER_Y | 0.02 | 0.01 | 0.01 | 0.46 | 0.18 | **0.55** | 0.65 |

Table 22: Privateeye fine-grained results. Breakdown of F1 Scores for every state variable in Privateeye for every method for probes where data was collected by random agent

| METHOD | MAJ-CLF | RANDOM-CNN | VAE | PIXEL-PRED | CPC | ST-DIM | SUPERVISED |
|---|---|---|---|---|---|---|---|
| AGENT_X | 0.00 | 0.23 | 0.09 | 0.18 | 0.09 | **0.47** | 0.98 |
| AGENT_Y | 0.05 | 0.60 | 0.86 | **0.95** | 0.90 | 0.88 | 0.98 |
| CLOCK_0 | 0.28 | 0.88 | 0.91 | **1.00** | **1.00** | **1.00** | 1.00 |
| CLOCK_1 | 0.00 | 0.19 | 0.02 | 0.83 | **0.98** | 0.96 | 0.98 |
| DOVE_X | 0.22 | **1.00** | 0.99 | **1.00** | **1.00** | **1.00** | 0.98 |
| DOVE_Y | 0.42 | **1.00** | **1.00** | **1.00** | **1.00** | **1.00** | 1.00 |
| PLAYER_DIRECTION | 0.36 | 0.48 | 0.53 | 0.64 | 0.48 | **0.93** | 0.98 |
| ROOM_NUMBER | 0.16 | **1.00** | 0.99 | **1.00** | **1.00** | **1.00** | 1.00 |
| SCORE_0 | 0.63 | 0.93 | 0.88 | **1.00** | **1.00** | **1.00** | 0.87 |

Table 23: Qbert fine-grained results. Breakdown of F1 Scores for every state variable in Qbert for every method for probes where data was collected by random agent

| METHOD | MAJ-CLF | RANDOM-CNN | VAE | PIXEL-PRED | CPC | ST-DIM | SUPERVISED |
|---|---|---|---|---|---|---|---|
| GREEN_ENEMY_COLUMN | 0.53 | 0.75 | 0.75 | 0.69 | 0.82 | **0.88** | 0.83 |
| PLAYER_COLUMN | 0.06 | 0.38 | 0.38 | 0.47 | 0.59 | **0.62** | 0.74 |
| PLAYER_X | 0.11 | 0.27 | 0.25 | 0.46 | 0.55 | **0.65** | 0.71 |
| PLAYER_Y | 0.01 | 0.06 | 0.07 | 0.10 | 0.29 | **0.48** | 0.61 |

Table 24: Riverraid fine-grained results. Breakdown of F1 Scores for every state variable in Riverraid for every method for probes where data was collected by random agent

| METHOD | MAJ-CLF | RANDOM-CNN | VAE | PIXEL-PRED | CPC | ST-DIM | SUPERVISED |
|---|---|---|---|---|---|---|---|
| FUEL_METER_HIGH | 0.01 | 0.23 | 0.22 | **0.31** | **0.32** | 0.28 | 0.32 |
| FUEL_METER_LOW | 0.05 | 0.14 | **0.20** | 0.17 | **0.21** | 0.19 | 0.23 |
| MISSILE_X | 0.00 | **0.44** | 0.26 | 0.40 | 0.41 | 0.25 | 0.54 |
| MISSILE_Y | 0.15 | 0.22 | 0.24 | 0.51 | 0.42 | **0.74** | 0.97 |
| PLAYER_X | 0.02 | 0.52 | 0.33 | **0.55** | 0.51 | 0.34 | 0.70 |

Table 25: Seaquest fine-grained results. Breakdown of F1 Scores for every state variable in Seaquest for every method for probes where data was collected by random agent

| METHOD | MAJ-CLF | RANDOM-CNN | VAE | PIXEL-PRED | CPC | ST-DIM | SUPERVISED |
|---|---|---|---|---|---|---|---|
| DIVER_X_1 | 0.82 | 0.89 | 0.87 | 0.89 | 0.89 | **0.92** | 0.91 |
| DIVER_X_2 | 0.29 | 0.23 | 0.31 | 0.49 | **0.80** | **0.80** | 0.86 |
| DIVER_X_3 | 0.14 | 0.25 | 0.33 | 0.41 | **0.79** | 0.76 | 0.89 |
| ENEMY_OBSTACLE_X_0 | 0.06 | 0.19 | 0.13 | 0.28 | **0.44** | 0.42 | 0.50 |
| ENEMY_OBSTACLE_X_1 | 0.06 | 0.18 | 0.13 | 0.29 | **0.46** | 0.41 | 0.48 |
| ENEMY_OBSTACLE_X_2 | 0.06 | 0.17 | 0.13 | 0.28 | **0.45** | 0.43 | 0.46 |
| ENEMY_OBSTACLE_X_3 | 0.06 | 0.15 | 0.14 | 0.29 | **0.41** | 0.39 | 0.42 |
| MISSILE_DIRECTION | 0.26 | **0.54** | 0.51 | **0.55** | 0.53 | **0.55** | 0.76 |
| MISSILE_X | 0.48 | 0.56 | 0.55 | 0.53 | 0.56 | **0.60** | 0.90 |
| NUM_LIVES | 0.11 | 0.98 | **0.99** | **1.00** | **1.00** | **1.00** | 1.00 |
| OXYGEN_METER_VALUE | 0.05 | 0.65 | 0.38 | 0.69 | 0.86 | **0.91** | 0.91 |
| PLAYER_DIRECTION | 0.42 | 0.61 | 0.63 | **0.66** | 0.64 | 0.64 | 0.91 |
| PLAYER_X | 0.09 | 0.22 | 0.33 | 0.39 | **0.42** | 0.26 | 0.84 |
| PLAYER_Y | 0.24 | 0.86 | **0.90** | 0.87 | 0.73 | 0.81 | 0.93 |
| SCORE_1 | 0.28 | 0.82 | 0.85 | 0.79 | **0.99** | **0.99** | 1.00 |

Table 26: Spaceinvaders fine-grained results. Breakdown of F1 Scores for every state variable in Spaceinvaders for every method for probes where data was collected by random agent

| METHOD | MAJ-CLF | RANDOM-CNN | VAE | PIXEL-PRED | CPC | ST-DIM | SUPERVISED |
|---|---|---|---|---|---|---|---|
| ENEMIES_X | 0.00 | 0.91 | 0.85 | **0.97** | 0.95 | 0.91 | 0.99 |
| ENEMIES_Y | 0.45 | **0.97** | 0.89 | 0.84 | 0.92 | 0.84 | 0.91 |
| INVADERS_LEFT_COUNT | 0.02 | 0.34 | 0.58 | **0.61** | 0.58 | **0.60** | 0.76 |
| MISSILES_Y | 0.07 | 0.11 | 0.12 | 0.10 | 0.15 | **0.19** | 0.43 |
| NUM_LIVES | 0.63 | 0.70 | 0.66 | 0.72 | **0.78** | **0.77** | 0.81 |
| PLAYER_SCORE | 0.04 | 0.29 | 0.54 | 0.61 | 0.43 | **0.73** | 0.89 |
| PLAYER_X | 0.05 | 0.16 | 0.46 | **0.59** | 0.44 | 0.45 | 0.76 |

Table 27: Tennis fine-grained results. Breakdown of F1 Scores for every state variable in Tennis for every method for probes where data was collected by random agent

| METHOD | MAJ-CLF | RANDOM-CNN | VAE | PIXEL-PRED | CPC | ST-DIM | SUPERVISED |
|---|---|---|---|---|---|---|---|
| BALL_X | 0.24 | 0.29 | 0.29 | 0.48 | 0.61 | **0.63** | 0.79 |
| BALL_Y | 0.04 | 0.05 | 0.10 | 0.18 | **0.27** | **0.26** | 0.32 |
| ENEMY_SCORE | 0.11 | 0.95 | 0.53 | **1.00** | **1.00** | **1.00** | 1.00 |
| ENEMY_X | 0.08 | 0.23 | 0.27 | **0.56** | 0.53 | 0.43 | 0.89 |
| ENEMY_Y | 0.02 | 0.27 | 0.19 | 0.43 | 0.49 | **0.52** | 0.78 |
| PLAYER_X | 0.06 | 0.33 | 0.26 | **0.54** | 0.50 | 0.52 | 0.90 |
| PLAYER_Y | 0.02 | 0.25 | 0.16 | 0.37 | 0.40 | **0.43** | 0.79 |

Table 28: Venture fine-grained results. Breakdown of F1 Scores for every state variable in Venture for every method for probes where data was collected by random agent

| METHOD | MAJ-CLF | RANDOM-CNN | VAE | PIXEL-PRED | CPC | ST-DIM | SUPERVISED |
|---|---|---|---|---|---|---|---|
| NUM_LIVES | 0.21 | 0.81 | **0.99** | **1.00** | **1.00** | **0.99** | 1.00 |
| PLAYER_X | 0.00 | 0.02 | 0.03 | 0.07 | 0.11 | **0.32** | 0.43 |
| PLAYER_Y | 0.01 | 0.10 | 0.10 | 0.07 | 0.17 | **0.38** | 0.54 |
| SPRITE0_X | 0.00 | 0.05 | 0.05 | 0.22 | **0.29** | **0.29** | 0.48 |
| SPRITE0_Y | 0.18 | 0.31 | 0.16 | 0.29 | 0.29 | **0.39** | 0.59 |
| SPRITE1_X | 0.00 | 0.17 | 0.02 | 0.24 | **0.39** | 0.35 | 0.48 |
| SPRITE1_Y | 0.02 | 0.26 | 0.06 | 0.29 | **0.37** | **0.37** | 0.60 |
| SPRITE2_X | 0.03 | 0.17 | 0.09 | 0.38 | **0.51** | 0.49 | 0.70 |
| SPRITE2_Y | 0.04 | 0.32 | 0.14 | 0.53 | **0.68** | 0.59 | 0.79 |
| SPRITE3_X | 0.02 | 0.05 | 0.07 | 0.12 | **0.38** | 0.32 | 0.37 |
| SPRITE3_Y | 0.03 | 0.11 | 0.15 | 0.28 | **0.39** | 0.35 | 0.48 |
| SPRITE4_X | 0.00 | 0.26 | 0.01 | 0.25 | 0.28 | **0.30** | 0.34 |
| SPRITE4_Y | 0.04 | 0.32 | 0.09 | 0.40 | **0.42** | 0.40 | 0.53 |
| SPRITE5_X | 0.11 | 0.31 | 0.20 | 0.24 | 0.38 | **0.47** | 0.64 |
| SPRITE5_Y | 0.05 | 0.09 | 0.06 | 0.32 | 0.49 | **0.52** | 0.55 |

Table 29: Videopinball fine-grained results. Breakdown of F1 Scores for every state variable in Videopinball for every method for probes where data was collected by random agent

| METHOD | MAJ-CLF | RANDOM-CNN | VAE | PIXEL-PRED | CPC | ST-DIM | SUPERVISED |
|---|---|---|---|---|---|---|---|
| BALL_X | 0.00 | 0.01 | 0.01 | 0.01 | 0.08 | **0.21** | 0.64 |
| BALL_Y | 0.00 | 0.00 | 0.01 | 0.01 | 0.06 | **0.15** | 0.38 |
| PLAYER_LEFT_PADDLE_Y | 0.27 | **0.99** | 0.97 | **0.99** | 0.96 | **0.98** | 1.00 |
| PLAYER_RIGHT_PADDLE_Y | 0.25 | **0.99** | 0.98 | **0.99** | 0.98 | **0.98** | 1.00 |
| SCORE_1 | 0.00 | 0.09 | 0.38 | **0.66** | **0.66** | 0.65 | 0.95 |
| SCORE_2 | 0.00 | 0.13 | 0.37 | **0.76** | **0.76** | 0.71 | 0.95 |

Table 30: Yarsrevenge fine-grained results. Breakdown of F1 Scores for every state variable in Yarsrevenge for every method for probes where data was collected by random agent

| METHOD | MAJ-CLF | RANDOM-CNN | VAE | PIXEL-PRED | CPC | ST-DIM | SUPERVISED |
|---|---|---|---|---|---|---|---|
| ENEMY_QOTILE_Y | 0.00 | 0.40 | 0.16 | 0.41 | 0.72 | **0.80** | 0.93 |
| PLAYER_YAR_X | 0.02 | 0.22 | 0.04 | 0.08 | **0.35** | 0.33 | 0.90 |
| PLAYER_YAR_Y | 0.00 | 0.14 | 0.02 | 0.05 | **0.29** | 0.18 | 0.93 |
| QOTILE_MISSILE_X | 0.00 | 0.20 | 0.13 | 0.19 | 0.32 | **0.41** | 0.40 |
| QOTILE_MISSILE_Y | 0.00 | 0.06 | 0.04 | 0.06 | 0.09 | **0.14** | 0.50 |
| YAR_MISSILE_X | 0.01 | 0.04 | 0.02 | 0.04 | 0.08 | **0.14** | 0.37 |
| YAR_MISSILE_Y | 0.01 | 0.02 | 0.03 | 0.03 | 0.04 | **0.06** | 0.19 |

Table 31: Asteroids fine-grained results. Breakdown of F1 Scores for every state variable in Asteroids for every method for probes where data was collected by a pretrained PPO agent that was trained for 50M frames

| METHOD | MAJ-CLF | RANDOM-CNN | VAE | PIXEL-PRED | CPC | ST-DIM | SUPERVISED |
|---|---|---|---|---|---|---|---|
| PLAYER_X | 0.03 | 0.09 | 0.14 | 0.13 | 0.17 | **0.21** | 0.35 |
| PLAYER_Y | 0.20 | 0.30 | 0.27 | 0.32 | 0.32 | **0.41** | 0.94 |
| PLAYER_SCORE_LOW | 0.07 | 0.22 | 0.34 | 0.19 | 0.50 | **0.73** | 0.91 |
| PLAYER_MISSILE_X1 | 0.02 | 0.07 | 0.11 | 0.10 | 0.12 | **0.14** | 0.20 |
| PLAYER_MISSILE_X2 | 0.02 | 0.06 | 0.10 | 0.08 | 0.10 | **0.12** | 0.23 |
| PLAYER_MISSILE_Y1 | 0.58 | 0.59 | **0.65** | **0.64** | 0.59 | 0.60 | 0.68 |
| PLAYER_MISSILE_Y2 | 0.40 | 0.44 | **0.46** | **0.46** | 0.42 | 0.44 | 0.61 |
| PLAYER_SCORE_HIGH | 0.39 | 0.75 | **0.99** | 0.78 | 0.86 | **1.00** | 1.00 |
| ENEMY_ASTEROIDS_X_0 | 0.03 | **0.28** | **0.28** | 0.08 | 0.23 | 0.19 | 0.35 |
| ENEMY_ASTEROIDS_X_1 | 0.15 | **0.32** | 0.23 | 0.10 | **0.31** | 0.24 | 0.28 |
| ENEMY_ASTEROIDS_X_2 | 0.36 | 0.42 | 0.26 | 0.22 | **0.44** | 0.23 | 0.31 |
| ENEMY_ASTEROIDS_X_3 | **0.67** | 0.61 | 0.45 | 0.46 | 0.53 | 0.45 | 0.53 |
| ENEMY_ASTEROIDS_X_7 | 0.00 | 0.18 | 0.18 | 0.05 | **0.23** | 0.14 | 0.44 |
| ENEMY_ASTEROIDS_X_8 | 0.00 | 0.10 | **0.14** | 0.02 | **0.13** | 0.08 | 0.22 |
| ENEMY_ASTEROIDS_X_9 | 0.00 | 0.07 | 0.11 | 0.01 | **0.15** | 0.09 | 0.27 |
| ENEMY_ASTEROIDS_Y_0 | 0.03 | 0.17 | 0.23 | 0.04 | **0.40** | 0.19 | 0.91 |
| ENEMY_ASTEROIDS_Y_1 | 0.15 | 0.27 | 0.22 | 0.05 | **0.34** | 0.27 | 0.70 |
| ENEMY_ASTEROIDS_Y_2 | 0.36 | 0.37 | 0.24 | 0.22 | **0.45** | 0.23 | 0.62 |
| ENEMY_ASTEROIDS_Y_3 | **0.67** | 0.61 | 0.42 | 0.46 | 0.58 | 0.47 | 0.65 |
| ENEMY_ASTEROIDS_Y_5 | 0.56 | 0.51 | **0.74** | 0.59 | 0.61 | 0.63 | 0.75 |
| ENEMY_ASTEROIDS_Y_7 | 0.06 | 0.16 | 0.22 | 0.04 | **0.40** | 0.25 | 0.93 |
| ENEMY_ASTEROIDS_Y_8 | 0.26 | 0.20 | 0.28 | 0.08 | **0.38** | 0.36 | 0.79 |
| ENEMY_ASTEROIDS_Y_9 | 0.36 | 0.28 | 0.37 | 0.19 | 0.41 | **0.49** | 0.74 |
| NUM_LIVES_DIRECTION | 0.00 | 0.06 | 0.12 | 0.08 | **0.17** | **0.16** | 0.19 |
| ENEMY_ASTEROIDS_X_10 | 0.03 | 0.10 | 0.12 | 0.02 | **0.16** | 0.08 | 0.22 |
| ENEMY_ASTEROIDS_X_11 | 0.00 | 0.06 | 0.08 | 0.01 | **0.15** | 0.08 | 0.17 |
| ENEMY_ASTEROIDS_X_12 | 0.02 | 0.07 | **0.35** | 0.04 | 0.14 | 0.19 | 0.29 |
| ENEMY_ASTEROIDS_Y_10 | 0.63 | 0.45 | **0.65** | 0.35 | 0.55 | 0.61 | 0.77 |
| ENEMY_ASTEROIDS_Y_12 | **0.86** | 0.77 | 0.54 | 0.81 | 0.75 | 0.66 | 0.85 |
| PLAYER_MISSILE1_DIRECTION | 0.58 | 0.60 | **0.65** | **0.65** | 0.59 | 0.61 | 0.66 |
| PLAYER_MISSILE2_DIRECTION | 0.41 | 0.45 | **0.47** | **0.46** | 0.43 | 0.44 | 0.53 |

Table 32: Berzerk fine-grained results. Breakdown of F1 Scores for every state variable in Berzerk for every method for probes where data was collected by a pretrained PPO agent that was trained for 50M frames

| METHOD | MAJ-CLF | RANDOM-CNN | VAE | PIXEL-PRED | CPC | ST-DIM | SUPERVISED |
|---|---|---|---|---|---|---|---|
| PLAYER_X | 0.00 | 0.20 | **0.33** | 0.31 | 0.30 | **0.32** | 0.74 |
| PLAYER_Y | 0.00 | 0.10 | 0.09 | 0.18 | 0.21 | **0.23** | 0.60 |
| NUM_LIVES | 0.05 | **0.63** | 0.48 | 0.61 | 0.59 | **0.62** | 0.67 |
| PLAYER_DIRECTION | 0.04 | 0.25 | 0.26 | **0.28** | 0.28 | **0.29** | 0.37 |
| GAME_LEVEL | 0.07 | 0.48 | 0.47 | 0.54 | **0.58** | **0.58** | 0.55 |
| PLAYER_SCORE_1 | 0.01 | 0.36 | 0.37 | 0.42 | 0.41 | **0.59** | 0.92 |
| PLAYER_SCORE_2 | 0.32 | 0.64 | 0.63 | **0.77** | 0.57 | **0.76** | 0.92 |
| ROBOT_MISSILE_X | 0.03 | 0.10 | **0.22** | 0.19 | 0.17 | 0.18 | 0.20 |
| ROBOT_MISSILE_Y | 0.03 | 0.18 | **0.24** | 0.22 | 0.18 | **0.23** | 0.32 |
| ENEMY_EVILOTTO_X | 0.21 | 0.17 | 0.32 | **0.67** | 0.38 | 0.45 | 0.55 |
| ENEMY_EVILOTTO_Y | 0.08 | **0.61** | 0.55 | **0.62** | 0.56 | **0.62** | 0.59 |
| ENEMY_ROBOTS_X_0 | 0.01 | 0.18 | 0.25 | **0.31** | 0.29 | 0.26 | 0.73 |
| ENEMY_ROBOTS_X_1 | 0.24 | 0.30 | 0.26 | 0.25 | **0.33** | **0.34** | 0.44 |
| ENEMY_ROBOTS_X_2 | 0.44 | 0.42 | 0.42 | 0.39 | 0.43 | **0.46** | 0.51 |
| ENEMY_ROBOTS_X_3 | 0.61 | 0.59 | 0.62 | 0.57 | 0.60 | **0.69** | 0.58 |
| ENEMY_ROBOTS_X_4 | 0.79 | 0.80 | 0.72 | 0.69 | 0.70 | **0.82** | 0.80 |
| ENEMY_ROBOTS_Y_0 | 0.16 | 0.38 | 0.59 | **0.77** | 0.66 | 0.67 | 0.96 |
| ENEMY_ROBOTS_Y_1 | 0.24 | 0.45 | 0.49 | 0.51 | **0.61** | 0.53 | 0.77 |
| ENEMY_ROBOTS_Y_2 | 0.44 | 0.56 | 0.54 | 0.55 | **0.62** | **0.62** | 0.71 |
| ENEMY_ROBOTS_Y_3 | 0.61 | 0.69 | 0.66 | 0.68 | 0.67 | **0.79** | 0.76 |
| ENEMY_ROBOTS_Y_4 | 0.79 | 0.82 | 0.77 | 0.78 | 0.77 | **0.85** | 0.83 |
| PLAYER_MISSILE_X | 0.00 | 0.04 | 0.04 | 0.06 | 0.05 | **0.07** | 0.42 |
| PLAYER_MISSILE_Y | 0.00 | 0.04 | 0.04 | **0.07** | 0.07 | **0.08** | 0.46 |
| ROBOTS_KILLED_COUNT | 0.05 | 0.39 | 0.33 | 0.42 | 0.43 | **0.46** | 0.51 |
| ROBOT_MISSILE_DIRECTION | 0.10 | 0.48 | 0.54 | 0.49 | **0.61** | **0.61** | 0.60 |
| PLAYER_MISSILE_DIRECTION | 0.05 | 0.33 | 0.34 | 0.39 | 0.34 | **0.46** | 0.55 |

Table 33: Bowling fine-grained results. Breakdown of F1 Scores for every state variable in Bowling for every method for probes where data was collected by a pretrained PPO agent that was trained for 50M frames

| METHOD | MAJ-CLF | RANDOM-CNN | VAE | PIXEL-PRED | CPC | ST-DIM | SUPERVISED |
|---|---|---|---|---|---|---|---|
| PLAYER_Y | 0.38 | 0.80 | 0.28 | 0.88 | 0.91 | **1.00** | 1.00 |
| BALL_X | 0.00 | 0.02 | 0.00 | 0.18 | 0.55 | **0.86** | 0.87 |
| BALL_Y | 0.43 | 0.44 | 0.37 | 0.63 | 0.87 | **0.98** | 1.00 |
| SCORE | 0.00 | 0.34 | 0.22 | 0.96 | 0.94 | **1.00** | 1.00 |
| PIN_EXISTENCE_3 | 0.24 | 0.84 | 0.97 | 0.96 | **0.99** | 1.00 | 1.00 |
| PIN_EXISTENCE_6 | 0.45 | 0.76 | 0.93 | 0.96 | **0.98** | 0.99 | 1.00 |
| PIN_EXISTENCE_7 | 0.24 | 0.85 | 0.98 | 0.98 | **0.99** | 1.00 | 1.00 |
| FRAME_NUMBER_DISPLAY | 0.01 | 0.80 | **1.00** | **1.00** | **1.00** | **1.00** | 1.00 |

Table 34: Boxing fine-grained results. Breakdown of F1 Scores for every state variable in Boxing for every method for probes where data was collected by a pretrained PPO agent that was trained for 50M frames

| METHOD | MAJ-CLF | RANDOM-CNN | VAE | PIXEL-PRED | CPC | ST-DIM | SUPERVISED |
|---|---|---|---|---|---|---|---|
| PLAYER_X | 0.31 | 0.65 | 0.70 | **0.87** | 0.55 | 0.83 | 0.94 |
| PLAYER_Y | 0.00 | 0.37 | 0.37 | **0.45** | 0.13 | 0.33 | 0.75 |
| CLOCK | 0.00 | 0.08 | 0.03 | 0.53 | **0.93** | **0.94** | 0.97 |
| ENEMY_X | 0.00 | 0.45 | 0.39 | **0.64** | 0.18 | **0.63** | 0.85 |
| ENEMY_Y | 0.00 | 0.24 | 0.30 | **0.38** | 0.09 | 0.30 | 0.79 |
| ENEMY_SCORE | 0.00 | 0.05 | 0.17 | 0.77 | 0.42 | **0.97** | 0.90 |
| PLAYER_SCORE | 0.00 | 0.04 | 0.02 | 0.33 | 0.07 | **0.93** | 1.00 |

Table 35: Breakout fine-grained results. Breakdown of F1 Scores for every state variable in Breakout for every method for probes where data was collected by a pretrained PPO agent that was trained for 50M frames

| METHOD | MAJ-CLF | RANDOM-CNN | VAE | PIXEL-PRED | CPC | ST-DIM | SUPERVISED |
|---|---|---|---|---|---|---|---|
| PLAYER_X | 0.01 | 0.34 | 0.70 | 0.19 | 0.57 | **0.81** | 0.96 |
| BALL_X | 0.00 | 0.01 | 0.01 | 0.04 | 0.20 | **0.34** | 0.56 |
| BALL_Y | 0.00 | 0.01 | 0.01 | 0.02 | 0.19 | **0.27** | 0.52 |
| SCORE | 0.02 | 0.39 | 0.91 | **0.94** | 0.76 | 0.88 | 0.98 |
| BLOCK_BIT_MAP_0 | 0.11 | 0.49 | **0.72** | **0.73** | 0.71 | 0.71 | 1.00 |
| BLOCK_BIT_MAP_1 | 0.31 | 0.48 | **0.78** | 0.71 | 0.74 | 0.62 | 1.00 |
| BLOCK_BIT_MAP_6 | 0.06 | 0.46 | 0.56 | 0.56 | 0.58 | **0.61** | 1.00 |
| BLOCK_BIT_MAP_7 | 0.39 | 0.66 | 0.71 | **0.84** | 0.69 | 0.57 | 0.99 |
| BLOCK_BIT_MAP_12 | 0.42 | **0.92** | 0.75 | 0.76 | **0.91** | 0.70 | 1.00 |
| BLOCK_BIT_MAP_13 | 0.52 | 0.84 | **0.90** | 0.87 | 0.78 | 0.74 | 1.00 |
| BLOCK_BIT_MAP_18 | 0.05 | 0.55 | 0.62 | **0.74** | 0.47 | 0.50 | 1.00 |
| BLOCK_BIT_MAP_19 | 0.19 | 0.68 | 0.62 | **0.76** | 0.74 | 0.52 | 0.98 |
| BLOCK_BIT_MAP_20 | 0.79 | 0.77 | **0.96** | 0.79 | 0.67 | 0.70 | 0.98 |
| BLOCK_BIT_MAP_24 | 0.07 | 0.55 | 0.59 | 0.61 | **0.65** | 0.52 | 0.95 |
| BLOCK_BIT_MAP_25 | 0.11 | 0.56 | **0.68** | 0.59 | 0.47 | 0.47 | 0.93 |
| BLOCK_BIT_MAP_26 | 0.47 | 0.65 | **0.81** | 0.77 | **0.80** | 0.75 | 0.92 |
| BLOCKS_HIT_COUNT | 0.00 | 0.34 | **0.74** | 0.63 | 0.51 | 0.69 | 0.87 |

Table 36: Demonattack fine-grained results. Breakdown of F1 Scores for every state variable in Demonattack for every method for probes where data was collected by a pretrained PPO agent that was trained for 50M frames

| METHOD | MAJ-CLF | RANDOM-CNN | VAE | PIXEL-PRED | CPC | ST-DIM | SUPERVISED |
|---|---|---|---|---|---|---|---|
| PLAYER_X | 0.00 | 0.11 | 0.01 | 0.14 | 0.08 | **0.42** | 0.82 |
| NUM_LIVES | 0.01 | 0.27 | 0.24 | 0.42 | **0.99** | **0.99** | 1.00 |
| LEVEL | 0.04 | 0.40 | 0.51 | 0.52 | 0.77 | **0.86** | 0.87 |
| ENEMY_X1 | 0.01 | 0.02 | 0.01 | 0.06 | **0.09** | 0.07 | 0.15 |
| ENEMY_X2 | 0.00 | 0.03 | 0.00 | **0.05** | **0.06** | **0.06** | 0.12 |
| ENEMY_X3 | 0.00 | 0.02 | 0.00 | **0.04** | **0.05** | 0.03 | 0.13 |
| ENEMY_Y1 | 0.00 | 0.06 | 0.02 | 0.07 | **0.10** | 0.09 | 0.20 |
| ENEMY_Y2 | 0.00 | 0.05 | 0.01 | 0.07 | **0.09** | 0.07 | 0.18 |
| ENEMY_Y3 | 0.00 | 0.06 | 0.02 | 0.08 | **0.12** | 0.09 | 0.27 |
| MISSILE_Y | 0.09 | 0.11 | 0.12 | 0.17 | 0.24 | **0.55** | 0.96 |

Table 37: Freeway fine-grained results. Breakdown of F1 Scores for every state variable in Freeway for every method for probes where data was collected by a pretrained PPO agent that was trained for 50M frames

| METHOD | MAJ-CLF | RANDOM-CNN | VAE | PIXEL-PRED | CPC | ST-DIM | SUPERVISED |
|---|---|---|---|---|---|---|---|
| PLAYER_Y | 0.01 | 0.01 | 0.01 | **0.59** | 0.06 | 0.31 | 0.87 |
| SCORE | 0.00 | 0.16 | 0.05 | 0.22 | 0.22 | **0.50** | 0.42 |
| ENEMY_CAR_X_0 | 0.00 | 0.88 | 0.00 | **1.00** | 0.97 | **1.00** | 1.00 |
| ENEMY_CAR_X_1 | 0.00 | 0.97 | 0.00 | **1.00** | 0.98 | **1.00** | 1.00 |
| ENEMY_CAR_X_2 | 0.00 | 0.96 | 0.00 | 0.99 | 0.87 | **1.00** | 1.00 |
| ENEMY_CAR_X_3 | 0.00 | 0.94 | 0.00 | 0.99 | 0.69 | **1.00** | 1.00 |
| ENEMY_CAR_X_4 | 0.00 | 0.92 | 0.00 | 0.98 | 0.84 | **1.00** | 1.00 |
| ENEMY_CAR_X_5 | 0.00 | 0.91 | 0.00 | 0.98 | 0.73 | **1.00** | 1.00 |
| ENEMY_CAR_X_6 | 0.00 | 0.93 | 0.00 | 0.98 | 0.62 | **1.00** | 1.00 |
| ENEMY_CAR_X_7 | 0.00 | 0.91 | 0.00 | 0.99 | 0.93 | **1.00** | 1.00 |
| ENEMY_CAR_X_8 | 0.00 | 0.88 | 0.00 | **1.00** | 0.92 | **1.00** | 1.00 |
| ENEMY_CAR_X_9 | 0.00 | 0.66 | 0.00 | 0.99 | 0.96 | **1.00** | 1.00 |

Table 38: Frostbite fine-grained results. Breakdown of F1 Scores for every state variable in Frostbite for every method for probes where data was collected by a pretrained PPO agent that was trained for 50M frames

| METHOD | MAJ-CLF | RANDOM-CNN | VAE | PIXEL-PRED | CPC | ST-DIM | SUPERVISED |
|---|---|---|---|---|---|---|---|
| PLAYER_X | 0.02 | 0.19 | 0.14 | **0.40** | 0.37 | 0.33 | 0.74 |
| PLAYER_Y | 0.02 | 0.50 | 0.29 | 0.66 | **0.71** | 0.66 | 0.87 |
| NUM_LIVES | 0.14 | 0.92 | 0.87 | **0.99** | **1.00** | **0.99** | 1.00 |
| PLAYER_DIRECTION | 0.03 | 0.12 | 0.12 | **0.15** | **0.16** | 0.14 | 0.23 |
| SCORE_1 | 0.30 | 0.93 | 0.89 | 0.95 | **1.00** | 0.98 | 1.00 |
| SCORE_2 | 0.03 | 0.43 | 0.32 | 0.81 | **0.96** | 0.70 | 1.00 |
| ENEMY_X_0 | 0.69 | 0.91 | 0.70 | 0.87 | 0.92 | **0.94** | 0.99 |
| ENEMY_X_1 | 0.24 | 0.67 | 0.34 | 0.79 | **0.90** | 0.87 | 0.99 |
| ENEMY_X_2 | 0.45 | 0.74 | 0.61 | 0.82 | **0.91** | 0.89 | 0.99 |
| ENEMY_X_3 | 0.08 | 0.77 | 0.30 | 0.77 | **0.89** | **0.88** | 0.99 |
| TOP_ROW_ICEFLOW_X | 0.01 | **0.92** | 0.55 | 0.84 | 0.90 | 0.88 | 0.99 |
| IGLOO_BLOCKS_COUNT | 0.25 | 0.58 | 0.67 | 0.87 | **0.93** | 0.81 | 0.98 |
| THIRD_ROW_ICEFLOW_X | 0.01 | **0.91** | 0.63 | 0.88 | **0.91** | **0.90** | 1.00 |
| FOURTH_ROW_ICEFLOW_X | 0.01 | **0.88** | 0.63 | 0.85 | **0.88** | 0.87 | 1.00 |
| SECOND_ROW_ICEFLOW_X | 0.00 | **0.94** | 0.56 | 0.90 | **0.94** | 0.91 | 0.99 |

Table 39: Hero fine-grained results. Breakdown of F1 Scores for every state variable in Hero for every method for probes where data was collected by a pretrained PPO agent that was trained for 50M frames

| METHOD | MAJ-CLF | RANDOM-CNN | VAE | PIXEL-PRED | CPC | ST-DIM | SUPERVISED |
|---|---|---|---|---|---|---|---|
| PLAYER_X | 0.00 | 0.41 | 0.55 | 0.63 | **0.66** | 0.53 | 0.94 |
| PLAYER_Y | 0.42 | 0.38 | 0.61 | 0.66 | **0.75** | **0.74** | 0.93 |
| SCORE_1 | 0.02 | 0.21 | 0.20 | 0.35 | **0.84** | 0.58 | 0.97 |
| SCORE_0 | 0.01 | 0.39 | 0.45 | 0.48 | **0.92** | 0.67 | 0.98 |
| POWER_METER | 0.01 | 0.50 | 0.47 | 0.52 | **0.87** | 0.79 | 0.92 |
| ROOM_NUMBER | 0.05 | **0.99** | 0.95 | **0.99** | **1.00** | 0.98 | 0.96 |
| LEVEL_NUMBER | 0.22 | 0.97 | 0.98 | **0.99** | **1.00** | **0.99** | 0.98 |
| DYNAMITE_COUNT | 0.12 | 0.59 | 0.64 | 0.89 | **1.00** | 0.97 | 1.00 |

Table 40: Montezumarevenge fine-grained results. Breakdown of F1 Scores for every state variable in Montezumarevenge for every method for probes where data was collected by a pretrained PPO agent that was trained for 50M frames

| METHOD | MAJ-CLF | RANDOM-CNN | VAE | PIXEL-PRED | CPC | ST-DIM | SUPERVISED |
|---|---|---|---|---|---|---|---|
| PLAYER_X | 0.32 | 0.65 | 0.59 | 0.70 | 0.70 | **0.75** | 0.92 |
| PLAYER_Y | 0.01 | 0.54 | 0.43 | **0.81** | 0.78 | **0.81** | 0.94 |
| NUM_LIVES | 0.04 | 0.98 | **0.99** | 0.94 | **1.00** | **1.00** | 1.00 |
| ROOM_STATE | 0.01 | 0.18 | 0.10 | 0.30 | 0.33 | **0.41** | 0.59 |
| ENEMY_SKULL_X | 0.00 | 0.72 | 0.51 | 0.69 | **0.77** | 0.71 | 0.81 |
| PLAYER_DIRECTION | 0.02 | 0.45 | 0.35 | 0.55 | **0.64** | 0.61 | 0.73 |
| ITEMS_IN_INVENTORY_COUNT | 0.32 | 0.58 | 0.54 | 0.60 | **0.69** | 0.67 | 0.97 |

Table 41: Mspacman fine-grained results. Breakdown of F1 Scores for every state variable in Mspacman for every method for probes where data was collected by a pretrained PPO agent that was trained for 50M frames

| METHOD | MAJ-CLF | RANDOM-CNN | VAE | PIXEL-PRED | CPC | ST-DIM | SUPERVISED |
|---|---|---|---|---|---|---|---|
| PLAYER_X | 0.01 | 0.19 | 0.20 | **0.42** | 0.35 | 0.34 | 0.69 |
| PLAYER_Y | 0.05 | 0.25 | 0.28 | 0.41 | **0.45** | 0.42 | 0.71 |
| NUM_LIVES | 0.13 | 0.94 | **1.00** | 0.94 | 0.97 | 0.91 | 1.00 |
| PLAYER_DIRECTION | 0.16 | 0.38 | 0.37 | **0.53** | 0.50 | 0.47 | 0.87 |
| PLAYER_SCORE | 0.01 | 0.38 | 0.38 | **0.97** | 0.35 | 0.91 | 1.00 |
| FRUIT_X | 0.42 | 0.65 | 0.69 | 0.63 | **0.73** | 0.71 | 0.74 |
| FRUIT_Y | 0.14 | 0.26 | 0.28 | 0.21 | **0.43** | 0.17 | 0.42 |
| ENEMY_SUE_Y | 0.01 | 0.22 | 0.24 | 0.34 | **0.37** | 0.34 | 0.39 |
| ENEMY_INKY_X | 0.02 | 0.18 | 0.19 | **0.32** | **0.32** | 0.29 | 0.41 |
| ENEMY_INKY_Y | 0.02 | 0.25 | 0.20 | 0.37 | **0.40** | 0.38 | 0.47 |
| ENEMY_PINKY_X | 0.02 | 0.21 | 0.20 | **0.33** | 0.31 | 0.31 | 0.44 |
| ENEMY_PINKY_Y | 0.03 | 0.23 | 0.24 | 0.35 | **0.38** | 0.34 | 0.43 |
| ENEMY_BLINKY_X | 0.01 | 0.16 | 0.15 | 0.27 | **0.30** | 0.25 | 0.36 |
| ENEMY_BLINKY_Y | 0.04 | 0.20 | 0.20 | **0.32** | **0.32** | **0.31** | 0.41 |
| DOTS_EATEN_COUNT | 0.00 | 0.05 | 0.12 | **0.21** | 0.19 | 0.16 | 0.46 |

Table 42: Pitfall fine-grained results. Breakdown of F1 Scores for every state variable in Pitfall for every method for probes where data was collected by a pretrained PPO agent that was trained for 50M frames

| METHOD | MAJ-CLF | RANDOM-CNN | VAE | PIXEL-PRED | CPC | ST-DIM | SUPERVISED |
|---|---|---|---|---|---|---|---|
| PLAYER_X | 0.01 | 0.03 | 0.07 | 0.13 | **0.41** | 0.36 | 0.82 |
| PLAYER_Y | 0.08 | 0.15 | 0.15 | 0.24 | 0.48 | **0.76** | 0.94 |
| ENEMY_LOGS_X | 0.77 | 0.97 | 0.89 | 0.97 | 0.97 | **0.99** | 1.00 |
| BOTTOM_OF_ROPE_Y | 0.05 | 0.24 | 0.12 | 0.70 | **0.89** | 0.81 | 0.89 |
| ENEMY_SCORPION_X | 0.01 | 0.88 | 0.90 | **0.96** | **0.95** | **0.95** | 0.99 |

Table 43: Pong fine-grained results. Breakdown of F1 Scores for every state variable in Pong for every method for probes where data was collected by a pretrained PPO agent that was trained for 50M frames

| METHOD | MAJ-CLF | RANDOM-CNN | VAE | PIXEL-PRED | CPC | ST-DIM | SUPERVISED |
|---|---|---|---|---|---|---|---|
| PLAYER_Y | 0.01 | 0.01 | 0.13 | 0.43 | 0.13 | **0.57** | 0.69 |
| ENEMY_Y | 0.00 | 0.26 | 0.00 | 0.66 | 0.67 | **0.82** | 0.92 |
| ENEMY_SCORE | 0.00 | 0.08 | 0.82 | **1.00** | 0.99 | **1.00** | 1.00 |
| PLAYER_SCORE | 0.02 | 0.06 | 0.71 | **1.00** | **1.00** | **1.00** | 1.00 |
| BALL_X | 0.01 | 0.02 | 0.03 | 0.55 | 0.66 | **0.70** | 0.81 |
| BALL_Y | 0.07 | 0.08 | 0.07 | 0.66 | 0.80 | **0.82** | 0.92 |

Table 44: Privateeye fine-grained results. Breakdown of F1 Scores for every state variable in Privateeye for every method for probes where data was collected by a pretrained PPO agent that was trained for 50M frames

| METHOD | MAJ-CLF | RANDOM-CNN | VAE | PIXEL-PRED | CPC | ST-DIM | SUPERVISED |
|---|---|---|---|---|---|---|---|
| PLAYER_DIRECTION | 0.36 | 0.53 | 0.72 | 0.71 | 0.61 | **0.92** | 0.99 |
| ROOM_NUMBER | 0.32 | **1.00** | **1.00** | **1.00** | **1.00** | **1.00** | 1.00 |
| DOVE_X | 0.32 | **1.00** | **1.00** | **1.00** | **1.00** | **1.00** | 1.00 |
| AGENT_X | 0.00 | 0.29 | 0.12 | 0.45 | 0.14 | **0.54** | 1.00 |
| AGENT_Y | 0.05 | 0.75 | 0.82 | **0.95** | 0.91 | 0.89 | 0.98 |
| CLOCK_0 | 0.28 | 0.94 | 0.82 | **1.00** | **1.00** | **1.00** | 1.00 |
| CLOCK_1 | 0.00 | 0.16 | 0.03 | 0.83 | **0.98** | 0.96 | 0.98 |

Table 45: Qbert fine-grained results. Breakdown of F1 Scores for every state variable in Qbert for every method for probes where data was collected by a pretrained PPO agent that was trained for 50M frames

| METHOD | MAJ-CLF | RANDOM-CNN | VAE | PIXEL-PRED | CPC | ST-DIM | SUPERVISED |
|---|---|---|---|---|---|---|---|
| PLAYER_X | 0.03 | 0.31 | 0.23 | 0.34 | **0.49** | **0.50** | 0.76 |
| PLAYER_Y | 0.02 | 0.09 | 0.11 | 0.11 | 0.25 | **0.29** | 0.66 |
| PLAYER_COLUMN | 0.05 | 0.34 | 0.33 | 0.36 | **0.54** | 0.52 | 0.69 |
| RED_ENEMY_COLUMN | 0.06 | 0.36 | 0.51 | 0.43 | **0.56** | 0.51 | 0.56 |
| GREEN_ENEMY_COLUMN | 0.13 | **0.60** | 0.55 | **0.60** | **0.61** | 0.51 | 0.62 |

Table 46: Riverraid fine-grained results. Breakdown of F1 Scores for every state variable in Riverraid for every method for probes where data was collected by a pretrained PPO agent that was trained for 50M frames

| METHOD | MAJ-CLF | RANDOM-CNN | VAE | PIXEL-PRED | CPC | ST-DIM | SUPERVISED |
|---|---|---|---|---|---|---|---|
| PLAYER_X | 0.01 | 0.41 | 0.27 | **0.47** | 0.37 | 0.20 | 0.76 |
| MISSILE_Y | 0.14 | 0.18 | 0.21 | 0.35 | **0.38** | **0.37** | 0.96 |
| MISSILE_X | 0.00 | 0.24 | 0.20 | **0.30** | 0.21 | 0.13 | 0.52 |
| FUEL_METER_LOW | 0.04 | 0.14 | 0.12 | **0.18** | 0.16 | **0.18** | 0.24 |
| FUEL_METER_HIGH | 0.01 | 0.12 | 0.21 | 0.27 | **0.35** | 0.23 | 0.32 |

Table 47: Seaquest fine-grained results. Breakdown of F1 Scores for every state variable in Seaquest for every method for probes where data was collected by a pretrained PPO agent that was trained for 50M frames

| METHOD | MAJ-CLF | RANDOM-CNN | VAE | PIXEL-PRED | CPC | ST-DIM | SUPERVISED |
|---|---|---|---|---|---|---|---|
| PLAYER_X | 0.45 | 0.78 | 0.77 | **0.86** | 0.67 | 0.64 | 0.95 |
| PLAYER_Y | 0.26 | 0.84 | **0.87** | **0.86** | 0.77 | 0.76 | 0.92 |
| NUM_LIVES | 0.13 | **1.00** | **1.00** | **1.00** | **1.00** | 0.99 | 1.00 |
| SCORE_1 | 0.08 | 0.73 | 0.45 | 0.97 | 0.74 | **0.99** | 1.00 |
| SCORE_0 | 0.02 | 0.69 | 0.59 | 0.96 | 0.95 | **0.98** | 1.00 |
| MISSILE_X | 0.42 | 0.42 | 0.46 | 0.66 | 0.48 | **0.71** | 0.92 |
| DIVER_X_0 | 0.73 | 0.80 | **0.83** | 0.79 | **0.82** | 0.80 | 0.98 |
| DIVER_X_1 | 0.58 | 0.64 | 0.75 | 0.63 | 0.58 | **0.81** | 0.94 |
| DIVER_X_2 | 0.20 | 0.23 | 0.22 | 0.46 | 0.60 | **0.64** | 0.89 |
| DIVER_X_3 | 0.18 | 0.24 | 0.22 | 0.49 | **0.63** | 0.62 | 0.91 |
| MISSILE_DIRECTION | 0.40 | 0.75 | 0.72 | **0.83** | 0.72 | 0.78 | 0.95 |
| ENEMY_OBSTACLE_X_0 | 0.01 | 0.06 | 0.04 | 0.12 | **0.18** | 0.14 | 0.21 |
| ENEMY_OBSTACLE_X_1 | 0.00 | 0.11 | 0.05 | 0.19 | **0.24** | 0.22 | 0.35 |
| ENEMY_OBSTACLE_X_2 | 0.00 | 0.21 | 0.04 | 0.27 | **0.46** | 0.41 | 0.64 |
| ENEMY_OBSTACLE_X_3 | 0.00 | 0.21 | 0.04 | 0.29 | **0.46** | 0.39 | 0.62 |
| OXYGEN_METER_VALUE | 0.00 | 0.74 | 0.28 | 0.79 | **0.91** | 0.88 | 0.95 |
| DIVERS_COLLECTED_COUNT | 0.43 | 0.96 | 0.85 | **1.00** | **1.00** | **1.00** | 1.00 |

Table 48: Spaceinvaders fine-grained results. Breakdown of F1 Scores for every state variable in Spaceinvaders for every method for probes where data was collected by a pretrained PPO agent that was trained for 50M frames

| METHOD | MAJ-CLF | RANDOM-CNN | VAE | PIXEL-PRED | CPC | ST-DIM | SUPERVISED |
|---|---|---|---|---|---|---|---|
| PLAYER_X | 0.00 | 0.27 | 0.47 | **0.54** | 0.37 | 0.43 | 0.84 |
| PLAYER_SCORE | 0.00 | 0.15 | 0.15 | 0.45 | 0.18 | **0.50** | 0.81 |
| ENEMIES_X | 0.00 | 0.72 | 0.63 | **0.79** | 0.74 | 0.72 | 0.97 |
| ENEMIES_Y | 0.10 | 0.48 | 0.45 | **0.53** | 0.46 | 0.49 | 0.50 |
| MISSILES_Y | 0.02 | 0.03 | 0.04 | 0.04 | 0.06 | **0.08** | 0.35 |
| INVADERS_LEFT_COUNT | 0.01 | 0.35 | 0.32 | 0.34 | 0.38 | **0.41** | 0.51 |

Table 49: Tennis fine-grained results. Breakdown of F1 Scores for every state variable in Tennis for every method for probes where data was collected by a pretrained PPO agent that was trained for 50M frames

| METHOD | MAJ-CLF | RANDOM-CNN | VAE | PIXEL-PRED | CPC | ST-DIM | SUPERVISED |
|---|---|---|---|---|---|---|---|
| PLAYER_X | 0.42 | 0.10 | 0.17 | **0.71** | 0.49 | 0.38 | 0.60 |
| BALL_Y | 0.04 | 0.04 | 0.08 | 0.21 | 0.06 | **0.28** | 0.17 |

Table 50: Venture fine-grained results. Breakdown of F1 Scores for every state variable in Venture for every method for probes where data was collected by a pretrained PPO agent that was trained for 50M frames

| METHOD | MAJ-CLF | RANDOM-CNN | VAE | PIXEL-PRED | CPC | ST-DIM | SUPERVISED |
|---|---|---|---|---|---|---|---|
| PLAYER_X | 0.00 | 0.01 | 0.01 | 0.01 | 0.13 | **0.33** | 0.43 |
| PLAYER_Y | 0.01 | 0.06 | 0.04 | 0.03 | 0.13 | **0.39** | 0.55 |
| NUM_LIVES | 0.10 | 0.79 | 0.94 | 0.97 | **1.00** | 0.97 | 1.00 |
| SPRITE0_X | 0.03 | 0.07 | 0.07 | 0.02 | 0.20 | **0.30** | 0.48 |
| SPRITE0_Y | 0.16 | 0.20 | 0.26 | 0.23 | **0.35** | 0.36 | 0.60 |
| SPRITE1_X | 0.01 | 0.07 | 0.07 | 0.06 | 0.22 | **0.34** | 0.53 |
| SPRITE1_Y | 0.04 | 0.18 | 0.07 | 0.13 | **0.34** | 0.31 | 0.59 |
| SPRITE2_X | 0.03 | 0.07 | 0.14 | 0.07 | **0.51** | 0.49 | 0.67 |
| SPRITE2_Y | 0.10 | 0.33 | 0.19 | 0.10 | 0.58 | **0.66** | 0.79 |
| SPRITE3_X | 0.04 | 0.07 | 0.06 | 0.07 | 0.26 | **0.36** | 0.42 |
| SPRITE3_Y | 0.07 | 0.13 | 0.14 | 0.11 | 0.30 | **0.40** | 0.55 |
| SPRITE4_X | 0.00 | 0.08 | 0.03 | 0.01 | 0.31 | **0.34** | 0.38 |
| SPRITE4_Y | 0.03 | 0.24 | 0.15 | 0.15 | **0.47** | **0.47** | 0.55 |
| SPRITE5_X | 0.13 | 0.21 | 0.23 | 0.33 | 0.40 | **0.60** | 0.67 |
| SPRITE5_Y | 0.07 | 0.10 | 0.07 | 0.04 | 0.45 | **0.51** | 0.55 |

Table 51: Videopinball fine-grained results. Breakdown of F1 Scores for every state variable in Videopinball for every method for probes where data was collected by a pretrained PPO agent that was trained for 50M frames

| METHOD | MAJ-CLF | RANDOM-CNN | VAE | PIXEL-PRED | CPC | ST-DIM | SUPERVISED |
|---|---|---|---|---|---|---|---|
| SCORE_1 | 0.00 | 0.09 | 0.34 | 0.59 | **0.65** | 0.46 | 0.92 |
| SCORE_2 | 0.00 | 0.10 | 0.25 | 0.72 | **0.74** | 0.53 | 0.87 |
| BALL_X | 0.00 | 0.01 | 0.02 | 0.05 | 0.05 | **0.20** | 0.60 |
| BALL_Y | 0.00 | 0.01 | 0.01 | 0.04 | 0.03 | **0.11** | 0.33 |
| PLAYER_LEFT_PADDLE_Y | 0.18 | **0.99** | 0.95 | **0.99** | 0.97 | **0.98** | 1.00 |
| PLAYER_RIGHT_PADDLE_Y | 0.59 | **0.99** | 0.98 | **1.00** | **0.99** | 0.97 | 1.00 |

Table 52: Yarsrevenge fine-grained results. Breakdown of F1 Scores for every state variable in Yarsrevenge for every method for probes where data was collected by a pretrained PPO agent that was trained for 50M frames

| METHOD | MAJ-CLF | RANDOM-CNN | VAE | PIXEL-PRED | CPC | ST-DIM | SUPERVISED |
|---|---|---|---|---|---|---|---|
| PLAYER_YAR_X | 0.16 | 0.27 | 0.19 | 0.18 | 0.46 | **0.50** | 0.94 |
| PLAYER_YAR_Y | 0.01 | 0.25 | 0.32 | 0.15 | **0.41** | 0.35 | 0.96 |
| YAR_MISSILE_X | 0.02 | 0.04 | 0.04 | 0.05 | 0.08 | **0.19** | 0.42 |
| YAR_MISSILE_Y | 0.01 | 0.05 | 0.08 | 0.06 | 0.09 | **0.14** | 0.38 |
| ENEMY_QOTILE_Y | 0.00 | 0.09 | 0.44 | 0.43 | 0.50 | **0.60** | 0.79 |
| QOTILE_MISSILE_X | 0.00 | 0.19 | 0.16 | 0.18 | **0.49** | 0.47 | 0.53 |
| QOTILE_MISSILE_Y | 0.00 | 0.05 | 0.05 | 0.07 | 0.14 | **0.25** | 0.57 |