[Reviews · NeurIPS 2019]

Reviewer 1



* Update after Author Response * I think Reviewer 2 brings up valid concerns about the quality of the benchmark dataset presented in the paper. In my view, the creation of this benchmark is the main contribution of this paper, so it is important that it has high value to future researchers. The authors respond to the concerns and made improvements to the dataset. In light of this, I still believe this is a good submission and worth accepting, but I am revising the score to 7. * Original Review * Overall this paper was very well written and a pleasure to read. The strongest part of the paper is the presentation of the new benchmark task as well as the experiments based on it. Unfortunately, the experiments and the benchmark are built on Atari games, which have little interest in the real-world; it would be nice to see attempts at different tasks.The proposed state representation approach doesn't appear to particularly novel, but it is helpful to see its performance as compared to other approaches. There is no theoretical work here, but the experiments appear to have been executed well, and the discussion of the results is illuminating. Overall, I think this paper constitutes a decent contribution to the literature.

Reviewer 2



## [AARI Benchmark] I have some concerns about design choices for building the benchmark. (1) Data Collection: I don’t think a random agent or PPO after 10M would be a good enough agent’s policy that would cover diverse enough / extensive portion of possible game states. Currently only first few phases/states of the game are considered. If it is going to be considered as a standard, well-established benchmark, it should be more comprehensive and complete (see examples below). For example, I think it is worthwhile making use of demonstration dataset (e.g. Hester et al. AAAI 2018 DQfD) because of their large coverage. - For example, one of the most important state information for Montezuma’s Revenge is (x, y location, room information, # of keys), as discussed in [Ecoffet et al. arXiv 2019; Go-Explore] [Tang et al. NIPS 2017; SmartHash]. The variables are only pertaining to the very first room of the game; in the later stages of the game the agent has to discover a lot more information (e.g. other enemies, other object items other than keys such as torch/sword), which is currently not discoverable but should be added later on. - Some variables can be meaningless or a “don’t care” value depending on context -- e.g. in Venture, depending which room we are at, some sprite x/y variables would be null values or of don’t-care values. How should the benchmark handle such cases? - It would also be helpful to provide a quick overview of data distribution for each variable in the dataset (min, max, # of, etc). (2) Explicitness metric: - Why only reporting F1 score? Justification of using F1 measure and detailed descriptions (i.e. how exactly it was calculated when it comes to multiclass) would be needed. Arguably, the F1 scores seems not intuitive to hint how accurate the prediction was. Why not prediction accuracy or recall, etc.? - For some variables (such as agent/object location, timing information) are continuous rather than simply multinomial. But it seems that the proposed evaluation metric does not take this continuity into account, as it is a 256-way classifier probing; so I think the evaluation metric would be quite sensitive. It would be really hard to be perfectly predict the RAM values, not allowing errors by few pixels. - (Suggestion) Regarding (x, y) location variables, it would be also interesting to introduce distance-based (e.g. Euclidean distance as in [Choi et al., ICLR 2019]) metrics with proper normalizations. For some other discrete variables over certain K number of possible values (rather than 256), it would be also interesting to study K-way classification performance. ## [Method, Evaluation and Experiments] - I think baseline methods are reasonably and well chosen (section 5.2). Extensive breakdown of per-game and per-variable metric report is very helpful. - Question: why N/A for “supervised” on Asteroids/Boxing, or on many entries in Appendix Table 3? - Baseline: for ablating local(spatial) information, we have “Global-T-DIM”. However, I think another baseline where temporal component is ablated (which would be very similar to [25]) should be also presented. - Empirically, it is not clear whether the method is really capturing important key state representations very well. To name a few specifically, for example, on Boxing (e.g. Table 4), the F-1 score on key state variables (x,y location) is not superior; Montezuma’s Revenge, Seaquest, Freeway (player x/y) scores are low. Overall the numbers are better than the baselines for many variables the improvement looks limited, with . - It might not be necessary for this paper, but it would be interesting to study whether learned representation or joint learning of ST-DIM and policy actually improve the performance of RL policy (e.g. integrated with DQN, A2C, PPO). Though the representation can encode some information about these state (which is decodable as done in this work), but telling which of them are really important information remains an open question. - Additional suggestions: Qualitative visualization (e.g. t-SNE embeddings) of learned representation would have greatly improved analysis and understanding of the method. ## [Minor/additional points] Atari settings: [height x width] is [210 x 160], not [160 x 210]. (Table 1 in appendix) L147: I think it is a gym interface, not ALE interface. In the appendix, it would be necessary to provide an example game frames/images for each F1 scores table to provide a better idea of what each variable denotes. ## [More Related work] I recommend to add more comprehensive discussion for learning representation for RL. Some suggestions are: - (Sermanet et al., 2017 “Time-Constrative Networks”) - (Choi et al., ICLR 2019 “Contingency-aware exploration”) -- Agent localization based on attention mechanism, on some ALE game environments. - (Warde-Farley, et al., ICLR 2019 “Unsupervised Control Through Non-Parametric Discriminative Rewards”) -- learning unsupervised representation from pixel images. - (Zhu et al., NeurIPS 2018 “Object-Oriented Dynamics Predictor”) -- learning object and dynamics representation in RL using action-conditional dynamics. - (Aytar et al., 2019 “Playing hard exploration games by watching YouTube”) - VIC (Kregor et al., 2016) can be also added to [23-26] (L29-30). - An Atari Model Zoo for Analyzing, Visualizing, and Comparing Deep Reinforcement Learning Agents (2018) ## [Concluding comments] Overall, I think this paper addresses an important problem and shows a good efforts towards a benchmark of state representation learning, but the level of significance and completeness of the approach and benchmark is not enough. Mostly the design of benchmark and results are questionable. I encourage the authors to improve the work more, and it can be an interesting and a good contribution to the community. ------------------------------------------------- ## Post-Rebuttal update: I appreciate the authors providing feedback and answering many questions. Having read the author feedback and other reviewers' comment, I am now inclined to recommend this paper towards acceptance (from 4->6), though only slightly above the borderline. I agree that this work has a good contribution of providing benchmark suite for state representation learning and that the authors have addressed the reviewer's concerns well. First, I had a concern about data collection — the authors have expanded the coverage of the game and added more entries in a few environments. I think it is now in better shape. That being said, it may still have some missing aspects of the entire state space (which is in fact impossible to include all) for especially hard-exploration games where the vanilla PPO agent will barely make enough progress, but this would be something that can be done or enhanced with in future works; this work is presenting a reasonably executed step towards state representation learning. Regarding experiments and metrics, I appreciate the authors improving discussions and providing more metrics. I still have some doubts about "non-ordinal classification" using standard CE on "non-categorical, ordered variables" (e.g. Agent Loc. / Score, Clock, etc.) — without any label smoothing or proper quantization — which seems less justified and questionable, but as the current choice of evaluation metric looks consistent and reasonable it would be fine overall. On method and experiments, it is also encouraging that the non-temporal baseline is added, which I think is important in demonstrating the effectiveness of ST-DIM. However, the authors should have reported the numbers in the rebuttal to make it more convincing. Also, I agree and think lack of analysis (as R3 mentioned) of the learned representation is another weakness of the paper. More minor comments: - It would be also great if the authors could provide a hint of how well these pre-trained PPO agents would perform in each of the game, e.g. by reporting an average score in the appendix. - Moreover, please make sure that the per-game breakdown of metric is included in the benchmark either as a part of the dataset released or as appendix of the final version of the paper. It should be based on the PPO agent as well as the random agent. - As R3 mentioned, please include a more description of background (i.e. DIM) to make it easier to understand and be self-contained.

Reviewer 3



Update: I have read the author's response and the other reviews, and I maintain my score -- I think this a good paper. While I still wish the analysis was a bit broader (as described in my review), I appreciate that the authors expanded their AARI methodology based on the suggestions of reviewer #2. --- Original review --- Originality: As mentioned above, InfoNCE-DIM was already described in detail in [25], the only addition this paper makes is using CPC-style linear prediction targets [24]. It does however seems to be performing quite well, so this combination is certainly worth documenting. Similarly, while the idea of using ALE ram state as ground truth is not new, there is value in creating an evaluation suite for representation learning. Clarity: The paper is well-written in general. The method section (3.1) is very short, to understand/re-implement the method it's necessary to read and understand DIM [25] and CPC [24] papers, but maybe that's fine as much of the focus is on evaluation. Quality/Significance: As self-supervised representation learning is becoming more popular again, it's important to have good tools for evaluating and inspecting these models, especially in contexts outside of image classification and vision. So I was actually quite excited to see much of the paper's focus being on AARI, and I encourage the authors to make the code and baselines included in the submission public and easy to use, should this paper get accepted. My main criticism with the paper is that the analysis and discussion presented in sections 5 and 6 is actually quite trivial. The main finding seems to be that generative models with an image reconstruction loss don't focus much on the small moving part of the screen which comprise much of the meaningful game state, while models with contrastive losses fare better in that aspect. This point is widely understood and easy to test even without AARI. On the other hand, there's a lot of interesting questions which would be more interesting to investigate, for example: - There are many self-supervised tasks with contrastive losses being suggested, from coloring, spatial arrangement, temporal prediction, etc. What are the strengths and failure modes regarding the learned representation? - Specifically, it's obvious why ST-DIM outperforms a VAE, but why is it better than CPC? Do the DIM patches make it less susceptible to distractors? And if yes, how can we show/test for this? - Does the AARI score translate into performance on downstream tasks ? Experiments which show this would strengthen the significance of the metric. Representation space which feel useful and semantically meaningful to human don't always translate into good performance, especially in RL. I encourage the authors to broaden the analysis with this type of questions.

[Author Response · NeurIPS 2019]

We thank all the reviewers for their thoughtful responses. As **R1** and **R3** note, we tried to focus on the benchmark and
evaluations within it. **R2** highlighted important questions about some design choices in the benchmark; we have made
active improvements on some of them, and better explained our rationale on others below. We have also expanded on
our discussion of ST-DIM in the draft, and included an additional figure to explain the contrastive task. We address
below some of the concerns that were raised.

**ST-DIM vs DIM (R1)**: DIM was originally introduced in the context of static
images. In this work, we extend DIM to work with temporal data by contrasting
local and global features across frames at different time steps instead of within the
same image.

**Real-World Setups (R1)**: Evaluating how well these algorithms transfer to the
real-world is an exciting direction. Since contrastive methods don't focus on pixel-
level details, we expect them to transfer better than generative methods. Moreover,
practical applications such as robotics are inherently spatio-temporal, which make
them conducive for methods like ST-DIM that exploit such structure.

**Coverage in AARI (R2)**: For multi-game rooms (MZR, Venture, Pitfall, Private-
Eye, Hero), we have expanded our RAM labels to include objects and enemies from all the rooms. For example, RAM
indices 44, 45 in MZR now correspond to: the key (in rooms 1,7,8,14), spider (rooms 4,13,21), torch (5), sword (6),
snake (9,11,22), jewel (0,10,15,23), amulet (19), bouncing skulls (2,3). These labels are now grouped by room, which
allows us to focus only on variables relevant in a room, and ignore other spurious/null values.

**Data Collection (R2)**: Our default data collection mode now also includes a PPO agent trained for 50M steps (results
below) over full observations, which significantly improves coverage over the 10M agent. We want to stress that the
default data collection modes are meant only for evaluating different representation algorithms, and we think 22 games
over multiple rooms provide adequate visual diversity to do so. Having said that, our RAM interface does expose labels
from all the rooms now, so exploration methods or new methods can still leverage them to make systematic evaluations.

|  | MAJ-CLF | RANDOM-CNN | VAE | PIXEL-PRED | CPC | ST-DIM | SUPERVISED |
|---|---|---|---|---|---|---|---|
| MEAN | 0.11 | 0.38 | 0.38 | 0.54 | 0.56 | **0.61** | 0.80 |

**N/A Results(R2)**: The missing values in Table 2 for supervised are: 0.60 for Asteroids, and 0.85 for Boxing.

**Related papers (R2)**: Thanks for pointing us to the relevant papers. We have now included them in Section 2 and they
have significantly improved our discussion of related work.

**Evaluation Metrics (R2)**: We want to stress that the output labels in our dataset are ordered, but not continuous (int8's).
Our use of cross-entropy for ordinal classification follows a variety of prior work that have used CE in lieu of MSE
regression [Colorful Colorization (Zhang 2016), DIM (Hjelm 2018, sec 4.3) etc.]. We use F1 score over accuracy
because the dataset has uneven label distribution (e.g. in Pitfall the class, "player y position" is more frequently 32
(ground level) than 21 (peak of a jump)). We have also added accuracy numbers to the appendix (summary results
below). Specifically, we use F1 score with average='weighted' from sklearn, this averages F1 score from each class
weighted by support. We will also add class distribution numbers (min, max, mode, etc.).

|  | MAJ-CLF | RANDOM-CNN | VAE | PIXEL-PRED | CPC | ST-DIM | SUPERVISED |
|---|---|---|---|---|---|---|---|
| MEAN | 0.24 | 0.48 | 0.43 | 0.60 | 0.62 | **0.69** | 0.82 |

**Non-temporal DIM baseline (R2)**: We ablated the temporal aspect in ST-DIM by removing any temporal cues from
the contrastive task, making the setup closer to DIM. We found it to perform much worse than ST-DIM indicating that
temporal cues are really important. We will add a game-by-game plot similar to other ablations in our updated draft.

**Open-source (R3):** Open-sourcing our code and benchmark is a top priority for us. In fact, since the submission,
cleaning up the code and improving the API have been our key focus. We are excited to make a public release soon!

**Interpretation of Results (R3):** Folk wisdom does say that contrastive methods should do better at small objects (than
generative models); however, we couldn't find such claims empirically verified. Moreover, we believe discussing
robustness to easy-to-exploit features is important as contrastive methods become more popular. Holistically, we
systematically compare contrastive and generative methods, insights from which should be helpful to the RL community
which has mostly focused on using generative methods such as VAEs for representation learning.

**ST-DIM's empirical superiority to CPC (R3):** ST-DIM performs better than CPC because it avoids the trap of
focusing on a single easily predictable factor [WDM (Ozair et. al, 2019)]. In Table 4, we see that both Global-T-DIM
and CPC focus only on easy factors like the clock and the score, but ST-DIM captures other factors as well, indicating
local objectives (the only difference b/w Global-T-DIM and ST-DIM) help make it robust to easy-to-exploit factors.

[Meta-Review · NeurIPS 2019]

The paper proposes a new state representation learning method and a new evaluation tool for learned representation. The results are interesting enough to be published.